# A Unified and General Framework for Continual Learning

**Zhenyi Wang[1], Yan Li[1], Li Shen[2], Heng Huang[1]**
[1]University of Maryland, College Park    [2]JD Explore Academy
{zwang169, yanli18, heng}@umd.edu,   mathshenli@gmail.com

## Abstract

Continual Learning (CL) focuses on learning from dynamic and changing data distributions while retaining previously acquired knowledge. Various methods have been developed to address the challenge of catastrophic forgetting, including regularization-based, Bayesian-based, and memory-replay-based techniques. However, these methods lack a unified framework and common terminology for describing their approaches. This research aims to bridge this gap by introducing a comprehensive and overarching framework that encompasses and reconciles these existing methodologies. Notably, this new framework is capable of encompassing established CL approaches as special instances within a unified and general optimization objective. An intriguing finding is that despite their diverse origins, these methods share common mathematical structures. This observation highlights the compatibility of these seemingly distinct techniques, revealing their interconnectedness through a shared underlying optimization objective. Moreover, the proposed general framework introduces an innovative concept called *refresh learning*, specifically designed to enhance the CL performance. This novel approach draws inspiration from neuroscience, where the human brain often sheds outdated information to improve the retention of crucial knowledge and facilitate the acquisition of new information. In essence, *refresh learning* operates by initially unlearning current data and subsequently relearning it. It serves as a versatile plug-in that seamlessly integrates with existing CL methods, offering an adaptable and effective enhancement to the learning process. Extensive experiments on CL benchmarks and theoretical analysis demonstrate the effectiveness of the proposed *refresh learning*.

## 1 Introduction

Continual learning (CL) is a dynamic learning paradigm that focuses on acquiring knowledge from data distributions that undergo continuous changes, thereby simulating real-world scenarios where new information emerges over time. The fundamental objective of CL is to adapt and improve a model's performance as it encounters new data while retaining the knowledge it has accumulated from past experiences. This pursuit, however, introduces a substantial challenge: the propensity to forget or overwrite previously acquired knowledge when learning new information. This phenomenon, known as catastrophic forgetting (McCloskey & Cohen, 1989), poses a significant hurdle in achieving effective CL. As a result, the development of strategies to mitigate the adverse effects of forgetting and enable harmonious integration of new and old knowledge stands as a critical and intricate challenge within the realm of CL research.

A plethora of approaches have been introduced to address the challenge of forgetting in CL. These methods span a range of strategies, encompassing Bayesian-based techniques (Nguyen et al., 2018; Kao et al., 2021), regularization-driven solutions (Kirkpatrick et al., 2017; Cha et al., 2021), and memory-replay-oriented methodologies (Riemer et al., 2019; Buzzega et al., 2020). These methods have been developed from distinct perspectives, but lacking a cohesive framework and a standardized terminology for their formulation.

In the present study, we endeavor to harmonize this diversity by casting these disparate categories of CL methods within a unified and general framework with the tool of Bregman divergence. As

Table 1: A unified framework for CL. We define a generalized CL optimization objective as $\mathcal{L}^{CL} = \mathcal{L}_{CE}(\boldsymbol{x}, y) + \alpha D_{\boldsymbol{\Phi}}(h_{\boldsymbol{\theta}}(\boldsymbol{x}), \boldsymbol{z}) + \beta D_{\boldsymbol{\Psi}}(\boldsymbol{\theta}, \boldsymbol{\theta}_{old})$. Where $\alpha \geq 0, \beta \geq 0$, $\mathcal{L}_{CE}(\boldsymbol{x}, y)$ is the loss function on new task, $D_{\boldsymbol{\Phi}}(h_{\boldsymbol{\theta}}(\boldsymbol{x}), \boldsymbol{z})$ is *output space* regularization represented as a Bregman divergence associated with function $\boldsymbol{\Phi}$, $D_{\boldsymbol{\Psi}}(\boldsymbol{\theta}, \boldsymbol{\theta}_{old})$ is *weight space* regularization represented as a Bregman divergence associated with function $\boldsymbol{\Psi}$. Several existing representative CL methods can be recovered from this general optimization objective by setting different $\boldsymbol{\Phi}$, $\boldsymbol{\Psi}$ and Bregman divergence.

| Category | Method | Ref | Recover Setting |
|---|---|---|---|
| Bayesian-based | VCL | Nguyen et al. (2018) | $\alpha = 0, \boldsymbol{\Psi}(p) = \int p(\boldsymbol{x}) \log p(\boldsymbol{x}) d\boldsymbol{x}$ |
| | NCL | Kao et al. (2021) | $\boldsymbol{\Phi}(\boldsymbol{p}) = \sum_{i=1}^{i=n} \boldsymbol{p}_i \log \boldsymbol{p}_i. \boldsymbol{\Psi} = \frac{1}{2} \|\boldsymbol{\theta}\|^2$ |
| Regularization-based | EWC | Kirkpatrick et al. (2017) | $\alpha = 0, \boldsymbol{\Psi}(\boldsymbol{\theta}) = \frac{1}{2} \boldsymbol{\theta}^T F \boldsymbol{\theta}$ |
| | CPR | Cha et al. (2021) | $\boldsymbol{\Phi}(\boldsymbol{p}) = \sum_{i=1}^{i=n} \boldsymbol{p}_i \log \boldsymbol{p}_i$ |
| Memory-replay-based | ER | Chaudhry et al. (2019b) | $\beta = 0, \boldsymbol{\Phi}(\boldsymbol{p}) = \sum_{i=1}^{i=n} \boldsymbol{p}_i \log \boldsymbol{p}_i$ |
| | DER | Buzzega et al. (2020) | $\beta = 0, \boldsymbol{\Phi}(\boldsymbol{x}) = \|\boldsymbol{x}\|^2$ |
| Novel CL method | `Refresh Learning` | Ours | Unlearn-relearn plug-in |

outlined in Table 1, we introduce a generalized CL optimization objective. Our framework is designed to flexibly accommodate this general objective, allowing for the recovery of a wide array of representative CL methods across different categories. This is achieved by configuring the framework according to specific settings corresponding to the desired CL approach. Through this unification, we uncover an intriguing revelation: while these methods ostensibly belong to different categories, they exhibit underlying mathematical structures that are remarkably similar. This revelation lays the foundation for a broader and more inclusive CL approach. Our findings have the potential to open avenues for the creation of a more generalized and effective framework for addressing the challenge of knowledge retention in CL scenarios.

Our unified CL framework offers insights into the limitations of existing CL methods. It becomes evident that current CL techniques predominantly address the forgetting issue by constraining model updates either in the output space or the model weight space. However, they tend to prioritize the preservation of existing knowledge while potentially neglecting the risk of over-memorization. Over-emphasizing the retention of existing knowledge doesn't necessarily lead to improved generalization, as the network's capacity may become occupied by outdated and less relevant information. This can impede the acquisition of new knowledge and the effective recall of pertinent old knowledge.

To address this issue, we propose a *refresh learning* mechanism with a first unlearning, then relearn the current loss function. This is inspired by two aspects. On one hand, forgetting can be beneficial for the human brain in various situations, as it helps in efficient information processing and decision-making (Davis & Zhong, 2017; Richards & Frankland, 2017; Gravitz, 2019; Wang et al., 2023b). One example is the phenomenon known as "cognitive load" (Sweller, 2011). Imagine a person navigating through a new big city for the first time. They encounter a multitude of new and potentially overwhelming information, such as street names, landmarks, and various details about the environment. If the brain were to retain all this information indefinitely, it could lead to cognitive overload, making it challenging to focus on important aspects and make decisions effectively. However, the ability to forget less relevant details allows the brain to prioritize and retain essential information. Over time, the person might remember key routes, important landmarks, and necessary information for future navigation, while discarding less critical details. This selective forgetting enables the brain to streamline the information it holds, making cognitive processes more efficient and effective. In this way, forgetting serves as a natural filter, helping individuals focus on the most pertinent information and adapt to new situations without being overwhelmed by an excess of irrelevant details. On the other hand, CL involves adapting to new tasks and acquiring new knowledge over time. If a model were to remember every detail from all previous tasks, it could quickly become impractical and resource-intensive. Forgetting less relevant information helps in managing memory resources efficiently, allowing the model to focus on the most pertinent knowledge (Feldman & Zhang, 2020). Furthermore, catastrophic interference occurs when learning new information disrupts previously learned knowledge. Forgetting less relevant details helps mitigate this interference, enabling the model to adapt to new tasks without severely impacting its performance on previously learned tasks. Our proposed *refresh learning* is designed as a straightforward plug-in, making it easily compatible with existing CL methods. Its seamless integration capability allows it to augment the performance of CL techniques, resulting in enhanced CL performance overall.

To illustrate the enhanced generalization capabilities of the proposed method, we conduct a comprehensive theoretical analysis. Our analysis demonstrates that *refresh learning* approximately minimizes

the Fisher Information Matrix (FIM) weighted gradient norm of the loss function. This optimization encourages the flattening of the loss landscape, ultimately resulting in improved generalization. Extensive experiments conducted on various representative datasets demonstrate the effectiveness of the proposed method. Our contributions are summarized as three-fold:

- We propose a generalized CL optimization framework that encompasses various CL approaches as special instances, including Bayesian-based, regularization-based, and memory-replay-based CL methods, which provides a new understanding of existing CL methods.
- Building upon our unified framework, we derive a new *refresh learning* mechanism with an unlearn-relearn scheme to more effectively combat the forgetting issue. The proposed method is a simple plug-in and can be seamlessly integrated with existing CL methods.
- We provide in-depth theoretical analysis to prove the generalization ability of the proposed *refresh learning* mechanism. Extensive experiments on several representative datasets demonstrate the effectiveness and efficiency of *refresh learning*.

## 2 RELATED WORK

**Continual Learning (CL)** (van de Ven et al., 2022) aims to learn non-stationary data distribution. Existing methods on CL can be classified into four classes. (1) Regularization-based methods regularize the model weights or model outputs to mitigate forgetting. Representative works include (Kirkpatrick et al., 2017; Zenke et al., 2017b; Chaudhry et al., 2018; Aljundi et al., 2018; Cha et al., 2021; Wang et al., 2021; Yang et al., 2023a). (2) Bayesian-based methods enforce model parameter posterior distributions not change much when learning new tasks. Representative works include (Nguyen et al., 2018; Kurle et al., 2019; Kao et al., 2021; Henning et al., 2021; Pan et al., 2020; Titsias et al., 2020; Rudner et al., 2022). (3) Memory-replay-based methods maintain a small memory buffer which stores a small number of examples from previous tasks and then replay later to mitigate forgetting. Representative works include (Lopez-Paz & Ranzato, 2017; Riemer et al., 2019; Chaudhry et al., 2019c; Buzzega et al., 2020; Pham et al., 2021; Arani et al., 2022; Caccia et al., 2022; Wang et al., 2022b;a; 2023c;a; Yang et al., 2023b). (4) Architecture-based methods dynamically update the networks or utilize subnetworks to mitigate forgetting. Representative works include (Mallya & Lazebnik, 2018; Serra et al., 2018; Li et al., 2019; Hung et al., 2019). Our work proposes a unified framework to encompass various CL methods as special cases and offers a new understanding of these CL methods.

**Machine Unlearning** (Guo et al., 2020; Wu et al., 2020; Bourtoule et al., 2021; Ullah et al., 2021) refers to the process of removing or erasing previously learned information or knowledge from a pre-trained model to comply with privacy regulations (Ginart et al., 2019). In contrast to existing approaches focused on machine unlearning, which seek to entirely eliminate data traces from pre-trained models, our *refresh learning* is designed to selectively and dynamically eliminate outdated or less relevant information from CL model. This selective unlearning approach enhances the ability of the CL model to better retain older knowledge while efficiently acquiring new task information.

## 3 PROPOSED FRAMEWORK AND METHOD

We present preliminary and problem setup in Section 3.1, our unified and general framework for CL in Section 3.2, and our proposed refresh learning which is built upon and derived from the proposed CL optimization framework in Section 3.3.

### 3.1 PRELIMINARY AND PROBLEM SETUP

**Continual Learning Setup** The standard CL problem involves learning a sequence of $N$ tasks, represented as $\mathcal{D}^{tr} = \{\mathcal{D}_1^{tr}, \mathcal{D}_2^{tr}, \cdots, \mathcal{D}_N^{tr}\}$. The training dataset $\mathcal{D}_k^{tr}$ for the $k^{th}$ task contains a collection of triplets: $(\boldsymbol{x}_i^k, y_i^k, \mathcal{T}_k)_{i=1}^{n_k}$, where $\boldsymbol{x}_i^k$ denotes the $i^{th}$ data example specific to task $k$, $y_i^k$ represents the associated data label for $\boldsymbol{x}_i^k$, and $\mathcal{T}_k$ is the task identifier. The primary objective is to train a neural network function, parameterized by $\boldsymbol{\theta}$, denoted as $g_{\boldsymbol{\theta}}(\boldsymbol{x})$. The goal is to achieve good performance on the test datasets from all the learned tasks, represented as $\mathcal{D}^{te} = \{\mathcal{D}_1^{te}, \mathcal{D}_2^{te}, \cdots, \mathcal{D}_N^{te}\}$, while ensuring that knowledge acquired from previous tasks is not forgotten.

**Bregman Divergence** Consider $\boldsymbol{\Phi}$: $\Omega \to \mathbb{R}$ as a strictly convex differentiable function and defined on a convex set $\Omega$. The Bregman divergence (Banerjee et al., 2005) related to $\boldsymbol{\Phi}$ for two points $\boldsymbol{p}$ and $\boldsymbol{q}$ within the set $\Omega$ can be understood as the discrepancy between the $\boldsymbol{\Phi}$ value at point $\boldsymbol{p}$ and the value obtained by approximating $\boldsymbol{\Phi}$ using first-order Taylor expansion at $\boldsymbol{q}$. It is defined as:

$$D_{\boldsymbol{\Phi}}(\boldsymbol{p}, \boldsymbol{q}) = \boldsymbol{\Phi}(\boldsymbol{p}) - \boldsymbol{\Phi}(\boldsymbol{q}) - \langle \nabla \boldsymbol{\Phi}(\boldsymbol{q}), \boldsymbol{p} - \boldsymbol{q} \rangle \tag{1}$$

where $\nabla \boldsymbol{\Phi}(\boldsymbol{q})$ is the gradient of $\boldsymbol{\Phi}$ at $\boldsymbol{q}$. $\langle, \rangle$ denotes the dot product between two vectors. In the upcoming section, we will employ Bregman divergence to construct a unified framework for CL.

## 3.2 A Unified and General Framework for CL

In this section, we reformulate several established CL algorithms in terms of a general optimization objective. Specifically, a more general CL optimization objective can be expressed as the following:

$$\mathcal{L}^{CL} = \underbrace{\mathcal{L}_{CE}(\boldsymbol{x}, y)}_{\text{new task}} + \alpha \underbrace{D_{\boldsymbol{\Phi}}(h_{\boldsymbol{\theta}}(\boldsymbol{x}), \boldsymbol{z})}_{\text{output space}} + \beta \underbrace{D_{\boldsymbol{\Psi}}(\boldsymbol{\theta}, \boldsymbol{\theta}_{old})}_{\text{weight space}} \tag{2}$$

where $\boldsymbol{\theta}$ denotes the CL model parameters. $\mathcal{L}_{CE}(\boldsymbol{x}, y)$ is the cross-entropy loss on the labeled data $(\boldsymbol{x}, y)$ for the current new task. $\alpha \geq 0, \beta \geq 0$. The term $D_{\boldsymbol{\Phi}}(h_{\boldsymbol{\theta}}(\boldsymbol{x}), \boldsymbol{z})$ represents a form of regularization in the *output space* of the CL model. It is expressed as the Bregman divergence associated with the function $\boldsymbol{\Phi}$. The constant vector $\boldsymbol{z}$ serves as a reference value and helps us prevent the model from forgetting previously learned tasks. Essentially, it is responsible for reducing changes in predictions for tasks the model has learned before. On the other hand, $D_{\boldsymbol{\Psi}}(\boldsymbol{\theta}, \boldsymbol{\theta}_{old})$ represents a form of regularization applied to the *weight space*. It is also expressed as a Bregman divergence, this time associated with the function $\boldsymbol{\Psi}$. The term $\boldsymbol{\theta}_{old}$ refers to the optimal model parameters that were learned for older tasks. It is used to ensure that the model doesn't adapt too rapidly to new tasks and prevent the model from forgetting the knowledge of earlier tasks. Importantly, these second and third terms in Eq. 2 work together to prevent forgetting of previously learned tasks. Additionally, it's worth noting that various existing CL methods can be seen as specific instances of this general framework we've described above. Specifically, we cast VCL (Nguyen et al., 2018), NCL (Kao et al., 2021), EWC (Kirkpatrick et al., 2017), CPR (Cha et al., 2021), ER (Chaudhry et al., 2019c) and DER (Buzzega et al., 2020) as special instances of the optimization objective, Eq. (2). Due to space constraints, we will only outline the essential steps for deriving different CL methods in the following. Detailed derivations can be found in Appendix A.

**ER as A Special Case** Experience replay (ER) (Riemer et al., 2019; Chaudhry et al., 2019c) is a memory-replay based method for mitigating forgetting in CL. We denote the network softmax output as $g_{\boldsymbol{\theta}}(\boldsymbol{x}) = softmax(u_{\boldsymbol{\theta}}(\boldsymbol{x}))$ and $\boldsymbol{y}$ as the one-hot vector for the ground truth label. We use $\mathbb{KL}$ to denote the KL-divergence between two probability distributions. We denote $\mathcal{M}$ as the memory buffer which stores a small amount of data from previously learned tasks. ER optimizes the objective:

$$\mathcal{L}^{CL} = \mathcal{L}_{CE}(\boldsymbol{x}, y) + \alpha \mathbb{E}_{(\boldsymbol{x}, y) \in \mathcal{M}} \mathcal{L}_{CE}(\boldsymbol{x}, y) \tag{3}$$

In this case, in Eq. (2), we set $\beta = 0$. We take $\boldsymbol{\Phi}$ to be the negative entropy function, i.e., $\boldsymbol{\Phi}(\boldsymbol{p}) = \sum_{i=1}^{i=n} \boldsymbol{p}_i \log \boldsymbol{p}_i$. We set $\boldsymbol{p} = g_{\boldsymbol{\theta}}(\boldsymbol{x})$, i.e., the softmax probability output of the neural network on the memory buffer data and $\boldsymbol{q}$ to be the one-hot vector of the ground truth class distribution. Then, $D_{\boldsymbol{\Phi}}(\boldsymbol{p}, \boldsymbol{q}) = \mathbb{KL}(g_{\boldsymbol{\theta}}(\boldsymbol{x}), \boldsymbol{y})$. We recovered the ER method.

**DER as A Special Case** DER (Buzzega et al., 2020) is a memory-replay based method. DER not only stores the raw memory samples, but also stores the network logits for memory buffer data examples. Specifically, it optimizes the following objective function:

$$\mathcal{L}^{CL} = \mathcal{L}_{CE}(\boldsymbol{x}, y) + \alpha \mathbb{E}_{(\boldsymbol{x}, y) \in \mathcal{M}} ||u_{\boldsymbol{\theta}}(\boldsymbol{x}) - \boldsymbol{z}||_2^2 \tag{4}$$

where $u_{\boldsymbol{\theta}}(\boldsymbol{x})$ is the network output logit before the softmax and $\boldsymbol{z}$ is the network output logit when storing the memory samples. In this case, in Eq. (2), we set $\beta = 0$. We take $\boldsymbol{\Phi}(\boldsymbol{x}) = ||\boldsymbol{x}||^2$. Then, we set $\boldsymbol{p} = u_{\boldsymbol{\theta}}(\boldsymbol{x})$ and $\boldsymbol{q} = \boldsymbol{z}$. Then, $D_{\boldsymbol{\Phi}}(\boldsymbol{p}, \boldsymbol{q}) = ||u_{\boldsymbol{\theta}}(\boldsymbol{x}) - \boldsymbol{z}||_2^2$. We recover the DER method.

**CPR as A Special Case** CPR (Cha et al., 2021) is a regularization-based method and adds an entropy regularization term to the CL model loss function. Specifically, it solves:

$$\mathcal{L}^{CL} = \mathcal{L}_{CE}(\boldsymbol{x}, y) - \alpha H(g_{\boldsymbol{\theta}}(\boldsymbol{x})) + \beta D_{\boldsymbol{\Psi}}(\boldsymbol{\theta}, \boldsymbol{\theta}_{old}) \tag{5}$$

Where $H(g_{\theta}(\boldsymbol{x}))$ is the entropy function on the classifier class probabilities output. In Eq. (2), we take $\boldsymbol{\Phi}$ to be the negative entropy function, i.e., $\boldsymbol{\Phi}(\boldsymbol{p}) = \sum_{i=1}^{i=n} \boldsymbol{p}_i \log \boldsymbol{p}_i$. We set $\boldsymbol{p} = g_{\theta}(\boldsymbol{x})$, i.e., the probability output of CL model on the current task data and $\boldsymbol{q} = \boldsymbol{v}$, i.e., the uniform distribution on the class probability distribution. For the third term, we can freely set any proper regularization on the weight space regularization. $D_{\boldsymbol{\Phi}}(\boldsymbol{p}, \boldsymbol{q}) = \mathbb{KL}(g_{\theta}(\boldsymbol{x}), \boldsymbol{v})$. We then recover the CPR method.

**EWC as A Special Case** Elastic Weight Consolidation (EWC) (Kirkpatrick et al., 2017), is a regularization-based technique. It achieves this by imposing a penalty on weight updates using the Fisher Information Matrix (FIM). The EWC can be expressed as the following objective:

$$\mathcal{L}^{CL} = \mathcal{L}_{CE}(\boldsymbol{x}, y) + \beta(\boldsymbol{\theta} - \boldsymbol{\theta}_{old})^T F(\boldsymbol{\theta} - \boldsymbol{\theta}_{old}) \tag{6}$$

where $\boldsymbol{\theta}_{old}$ is mean vector of the Gaussian Laplace approximation for previous tasks, $F$ is the diagonal of the FIM. In Eq. (2), we set $\alpha = 0$, we take $\boldsymbol{\Psi}(\boldsymbol{\theta}) = \frac{1}{2}\boldsymbol{\theta}^T F \boldsymbol{\theta}$. We set $\boldsymbol{p} = \boldsymbol{\theta}$ and $\boldsymbol{q} = \boldsymbol{\theta}_{old}$. $D_{\boldsymbol{\Psi}}(\boldsymbol{p}, \boldsymbol{q}) = (\boldsymbol{\theta} - \boldsymbol{\theta}_{old})^T F(\boldsymbol{\theta} - \boldsymbol{\theta}_{old})$. Then, we recover the EWC method.

**VCL as A Special Case** Variational continual learning (VCL) (Nguyen et al., 2018) is a Bayesian-based method for mitigating forgetting in CL. The basic idea of VCL is to constrain the current model parameter distribution to be close to that of previous tasks. It optimizes the following objective.

$$\mathcal{L}^{CL} = \mathcal{L}_{CE}(\boldsymbol{x}, y) + \beta\mathbb{KL}(P(\boldsymbol{\theta}|\mathcal{D}_{1:t}), P(\boldsymbol{\theta}_{old}|\mathcal{D}_{1:t-1})) \tag{7}$$

where $\mathcal{D}_{1:t}$ denotes the dataset from task 1 to $t$. $P(\boldsymbol{\theta}|\mathcal{D}_{1:t})$ is the posterior distribution of the model parameters on the entire task sequence $\mathcal{D}_{1:t}$. $P(\boldsymbol{\theta}_{old}|\mathcal{D}_{1:t-1})$ is the posterior distribution of the model parameters on the tasks $\mathcal{D}_{1:t-1}$. In this case, $P(\boldsymbol{\theta}|\mathcal{D}_{1:t})$ and $P(\boldsymbol{\theta}_{old}|\mathcal{D}_{1:t-1})$ are both continuous distributions. In this case, in Eq. (2), we set $\alpha = 0$. we take $\boldsymbol{\Psi}$ to be $\boldsymbol{\Psi}(p) = \int p(\boldsymbol{\theta}) \log p(\boldsymbol{\theta}) d\boldsymbol{\theta}$. We then set $p = P(\boldsymbol{\theta}|\mathcal{D}_{1:t})$ and $q = P(\boldsymbol{\theta}_{old}|\mathcal{D}_{1:t-1})$. We then recover the VCL method.

**Natural Gradient CL as A Special Case** Natural Gradient CL (Osawa et al., 2019; Kao et al., 2021) (NCL) is a Bayesian-based CL method. Specifically, NCL updates the CL model by the following damped (generalized to be more stable) natural gradient:

$$\boldsymbol{\theta}_{k+1} = \boldsymbol{\theta}_k - \eta(\alpha F + \beta I)^{-1}\nabla\mathcal{L}(\boldsymbol{\theta}) \tag{8}$$

where $F$ is the FIM for previous tasks, $I$ is the identity matrix and $\eta$ is the learning rate. For the second loss term in Eq. (2), we take $\boldsymbol{\Phi}$ to be the negative entropy function, i.e., $\boldsymbol{\Phi}(\boldsymbol{p}) = \sum_{i=1}^{i=n} \boldsymbol{p}_i \log \boldsymbol{p}_i$. For the third loss term in Eq. (2), we adopt the $\boldsymbol{\Psi}(\boldsymbol{\theta}) = \frac{1}{2}||\boldsymbol{\theta}||^2$. In Eq. (2), we employ the first-order Taylor expansion to approximate the second loss term and employ the second-order Taylor expansion to approximate the third loss term. We then recover the natural gradient CL method. Due to the space limitations, we put the detailed theoretical derivations in Appendix A.6.

## 3.3 REFRESH LEARNING AS A GENERAL PLUG-IN FOR CL

The above unified CL framework sheds light on the limitations inherent in current CL methodologies. It highlights that current CL methods primarily focus on addressing the problem of forgetting by limiting model updates in either the output space or the model weight space. However, these methods tend to prioritize preserving existing knowledge at the potential expense of neglecting the risk of over-memorization. Overemphasizing the retention of old knowledge may not necessarily improve generalization because it can lead to the network storing outdated and less relevant information, which can hinder acquiring new knowledge and recalling important older knowledge.

In this section, we propose a general and novel plug-in, called *refresh learning*, for existing CL methods to address the above-mentioned over-memorization. This approach involves a two-step process: first, unlearning on the current mini-batch to erase outdated and unimportant information contained in neural network weights, and then relearning the current loss function. The inspiration for this approach comes from two sources. Firstly, in human learning, the process of forgetting plays a significant role in acquiring new skills and recalling older knowledge, as highlighted in studies like (Gravitz, 2019; Wang et al., 2023b). This perspective aligns with findings in neuroscience (Richards & Frankland, 2017), where forgetting is seen as essential for cognitive processes, enhancing thinking abilities, facilitating decision-making, and improving learning effectiveness. Secondly, neural networks often tend to overly memorize outdated information, which limits their adaptability to learn new and relevant data while retaining older information. This is because their model capacity

becomes filled with irrelevant and unimportant data, impeding their flexibility in learning and recall, as discussed in (Feldman & Zhang, 2020).

Our *refresh learning* builds upon the unified framework developed in Section 3.2. Consequently, we obtain a class of novel CL methods to address the forgetting issue more effectively. It serves as a straightforward plug-in and can be seamlessly integrated with existing CL methods, enhancing the overall performance of CL techniques. We employ a probabilistic approach to account for uncertainty during the unlearning step. To do this, we denote the posterior distribution of the CL model parameter as $\rho(\boldsymbol{\theta}) \coloneqq P(\boldsymbol{\theta}|\mathcal{D})$, where $\coloneqq$ denotes a definition. This distribution is used to model the uncertainty that arises during the process of unlearning, specifically on the current mini-batch data $\mathcal{D}$.

The main objective is to minimize the KL divergence between the current CL model parameters posterior and the target unlearned model parameter posterior. We denote the CL model parameter posterior at time $t$ as $\rho_t$, the target unlearned posterior as $\mu$. The goal is to minimize $\mathbb{KL}(\rho_t||\mu)$. Following Wibisono (2018), we define the target unlearned posterior as a energy function $\mu = e^{-\omega}$ and $\omega = -\mathcal{L}^{CL}$. This KL divergence can be further decomposed as:

$$\mathbb{KL}(\rho_t||\mu) = \int \rho_t(\boldsymbol{\theta}) \log \frac{\rho_t(\boldsymbol{\theta})}{\mu(\boldsymbol{\theta})} d\boldsymbol{\theta} = -\int \rho_t(\boldsymbol{\theta}) \log \mu(\boldsymbol{\theta}) d\boldsymbol{\theta} + \int \rho_t(\boldsymbol{\theta}) \log \rho_t(\boldsymbol{\theta}) d\boldsymbol{\theta} \quad (9)$$

$$= H(\rho_t, \mu) - H(\rho_t)$$

where $H(\rho_t, \mu) \coloneqq -\mathbb{E}_{\rho_t} \log \mu$ is the cross-entropy between $\rho_t$ and $\mu$. $H(\rho_t) \coloneqq -\mathbb{E}_{\rho_t} \log \rho_t$ is the entropy of $\rho_t$. Then, we plug-in the above terms into Eq. (9), and obtain the following:

$$\mathbb{KL}(\rho_t||\mu) = -\mathbb{E}_{\rho_t} \log \mu + \mathbb{E}_{\rho_t} \log \rho_t = -\mathbb{E}_{\rho_t} \mathcal{L}^{CL} + \mathbb{E}_{\rho_t} \log \rho_t \quad (10)$$

The entire refresh learning includes both unlearning-relearning can be formulated as the following:

$$\min_{\boldsymbol{\theta}} \mathbb{E}_{\rho_{opt}} \mathcal{L}^{CL} \quad \text{(relearn)} \quad (11)$$

$$s.t. \ \rho_{opt} = \min_{\rho}[\mathcal{E}(\rho) = -\mathbb{E}_{\rho} \mathcal{L}^{CL} + \mathbb{E}_{\rho} \log \rho] \quad \text{(unlearn)} \quad (12)$$

where Eq. (12) is to unlearn on the current mini-batch by optimizing an energy functional in function space over the CL parameter posterior distributions. Given that the energy functional $\mathcal{E}(\rho)$, as defined in Eq. (12), represents the negative loss of $\mathcal{L}^{CL}$, it effectively promotes an increase in loss. Consequently, this encourages the unlearning of the current mini-batch data, steering it towards the desired target unlearned parameter distribution. After obtaining the optimal unlearned CL model parameter posterior distribution, $\rho_{opt}$, the CL model then relearns on the current mini-batch data by Eq. (11). However, Eq. (12) involves optimization within the probability distribution space, and it is typically challenging to find a solution directly. To address this challenge efficiently, we convert Eq. (12) into a Partial Differential Equation (PDE) as detailed below.

By Fokker-Planck equation (Kadanoff, 2000), gradient flow of KL divergence is as following:

$$\frac{\partial \rho_t}{\partial t} = div \left( \rho_t \nabla \frac{\delta \mathbb{KL}(\rho_t||\mu)}{\delta \rho}(\rho) \right) \quad (13)$$

$div \cdot (\boldsymbol{q}) \coloneqq \sum_{i=1}^{d} \partial_{\boldsymbol{z}^i} \boldsymbol{q}^i(\boldsymbol{z})$ is the divergence operator operated on a vector-valued function $\boldsymbol{q} : \mathbb{R}^d \to \mathbb{R}^d$, where $\boldsymbol{z}^i$ and $\boldsymbol{q}^i$ are the $i$ th element of $\boldsymbol{z}$ and $\boldsymbol{q}$. Then, since the first-variation of KL-divergence, i.e., $\frac{\delta \mathbb{KL}(\rho_t||\mu)}{\delta \rho}(\rho_t) = \log \frac{\rho_t}{\mu} + 1$ (Liu et al., 2022). We plug it into Eq. 13, and obtain the following:

$$\frac{\partial \rho_t(\boldsymbol{\theta})}{\partial t} = div(\rho_t(\boldsymbol{\theta}) \nabla (\log \frac{\rho_t(\boldsymbol{\theta})}{\mu} + 1)) = div(\nabla \rho_t(\boldsymbol{\theta}) + \rho_t(\boldsymbol{\theta}) \nabla \omega) \quad (14)$$

Then, (Ma et al., 2015) proposes a more general Fokker-Planck equation as following:

$$\frac{\partial \rho_t(\boldsymbol{\theta})}{\partial t} = div[([D(\boldsymbol{\theta}) + Q(\boldsymbol{\theta})])(\nabla \rho_t(\boldsymbol{\theta}) + \rho_t(\boldsymbol{\theta}) \nabla \omega)] \quad (15)$$

where $D(\boldsymbol{\theta})$ is a positive semidefinite matrix and $Q(\boldsymbol{\theta})$ is a skew-symmetric matrix. We plug in the defined $\omega = -\mathcal{L}^{CL}$ into the above equation, we can get the following PDE:

$$\frac{\partial \rho_t(\boldsymbol{\theta})}{\partial t} = div([D(\boldsymbol{\theta}) + Q(\boldsymbol{\theta})])[-\rho_t(\boldsymbol{\theta}) \nabla \mathcal{L}^{CL}(\boldsymbol{\theta}) + \nabla \rho_t(\boldsymbol{\theta})] \quad (16)$$

Intuitively, parameters that are less critical for previously learned tasks should undergo rapid unlearning to free up more model capacity, while parameters of higher importance should unlearn at a slower rate. This adaptive unlearning of vital parameters ensures that essential information is retained. To model this intuition, we set the matrix $D(\boldsymbol{\theta}) = F^{-1}$, where $F$ is the FIM on previous tasks and set $Q(\boldsymbol{\theta}) = \mathbf{0}$ (Patterson & Teh, 2013). Eq. (16) illustrates that the energy functional decreases along the steepest trajectory in probability distribution space to gradually unlearn the knowledge in current data. By discretizing Eq. (16), we can obtain the following parameter update equation:

$$\boldsymbol{\theta}^j = \boldsymbol{\theta}^{j-1} + \gamma[F^{-1}\nabla\mathcal{L}^{CL}(\boldsymbol{\theta}^{j-1})] + \mathcal{N}(0, 2\gamma F^{-1}) \tag{17}$$

where in Eq. (17), the precondition matrix $F^{-1}$ aims to regulate the unlearning process. Its purpose is to facilitate a slower update of important parameters related to previous tasks while allowing less critical parameters to update more rapidly. It's important to note that the Hessian matrix of KL divergence coincides with the FIM, which characterizes the local curvature of parameter changes. In practice, this relationship is expressed as $\mathbb{KL}(p(\boldsymbol{x}|\boldsymbol{\theta})|p(\boldsymbol{x}|\boldsymbol{\theta}+\boldsymbol{d})) \approx \frac{1}{2}\boldsymbol{d}^T F \boldsymbol{d}$. This equation identifies the steepest direction for achieving the fastest unlearning of the output

---

**Algorithm 1** Refresh Learning for General CL.

1: **REQUIRE:** model parameters $\boldsymbol{\theta}$, CL model learning rate $\eta$,
2: **for** $k = 1$ to $K$ **do** (number of CL steps)
3:    **for** $j = 1$ to $J$ **do** (unlearn steps)
4:       $\boldsymbol{\theta}_k^j = \boldsymbol{\theta}_k^{j-1} + \gamma[F^{-1}\nabla\mathcal{L}^{CL}(\boldsymbol{\theta}_k^{j-1})] + \mathcal{N}(0, 2\gamma F^{-1})$
5:    **end for**
6:    $\boldsymbol{\theta}_{k+1} = \boldsymbol{\theta}_k - \eta\nabla\mathcal{L}^{CL}(\boldsymbol{\theta}_k^j)$ (relearn step)
7: **end for**

---

probability distribution. To streamline computation and reduce complexity, we employ a diagonal approximation of the FIM. It is important to note that the FIM is only computed once after training one task, the overall computation cost of FIM is thus negligible. The parameter $\gamma$ represents the unlearning rate, influencing the pace of unlearning. Additionally, we introduce random noise $\mathcal{N}(0, 2\gamma F^{-1})$ to inject an element of randomness into the unlearning process, compelling it to thoroughly explore the entire posterior distribution rather than converging solely to a single point estimation.

**Refresh Learning As a Special Case** Now, we derive our *refresh learning* as a special case of Eq. 2:

$$\mathcal{L}_{unlearn} = \underbrace{\mathcal{L}_{CE}(\boldsymbol{x}, y) + 2\alpha D_{\boldsymbol{\Phi}}(h_{\boldsymbol{\theta}}(\boldsymbol{x}), \boldsymbol{z}) + \beta D_{\boldsymbol{\Psi}}(\boldsymbol{\theta}, \boldsymbol{\theta}_{old})}_{\mathcal{L}_{CL}} - \alpha D_{\boldsymbol{\Phi}}(h_{\boldsymbol{\theta}}(\boldsymbol{x}), \boldsymbol{z}) \tag{18}$$

In Eq. (18): we adopt the second-order Taylor expansion on $D_{\boldsymbol{\Phi}}(h_{\boldsymbol{\theta}}(\boldsymbol{x}), \boldsymbol{z})$ as the following:

$$D_{\boldsymbol{\Phi}}(h_{\boldsymbol{\theta}}(\boldsymbol{x}), \boldsymbol{z}) \approx D_{\boldsymbol{\Phi}}(h_{\boldsymbol{\theta}_k}(\boldsymbol{x}), \boldsymbol{z}) + \nabla_{\boldsymbol{\theta}} D_{\boldsymbol{\Phi}}(h_{\boldsymbol{\theta}_k}(\boldsymbol{x}), \boldsymbol{z})(\boldsymbol{\theta} - \boldsymbol{\theta}_k) + \frac{1}{2}(\boldsymbol{\theta} - \boldsymbol{\theta}_k)^T F(\boldsymbol{\theta} - \boldsymbol{\theta}_k) \tag{19}$$

Since $\nabla_{\boldsymbol{\theta}} D_{\boldsymbol{\Phi}}(h_{\boldsymbol{\theta}}(\boldsymbol{x}), \boldsymbol{z})$ is close to zero at the stationary point, i.e., $\boldsymbol{\theta}_k$, we thus only need to optimize the leading quadratic term in Eq. 19. we adopt the first-order Taylor expansion on $\mathcal{L}_{CL}$ as:

$$\mathcal{L}_{CL}(\boldsymbol{\theta}) \approx \mathcal{L}_{CL}(\boldsymbol{\theta}_k) + \nabla_{\boldsymbol{\theta}}\mathcal{L}_{CL}(\boldsymbol{\theta}_k)(\boldsymbol{\theta} - \boldsymbol{\theta}_k) \tag{20}$$

In summary, the approximate loss function for Eq. (18) can be expressed as the following:

$$\mathcal{L}_{unlearn} \approx \nabla_{\boldsymbol{\theta}}\mathcal{L}_{CL}(\boldsymbol{\theta}_k)(\boldsymbol{\theta} - \boldsymbol{\theta}_k) - \frac{\alpha}{2}(\boldsymbol{\theta} - \boldsymbol{\theta}_k)^T F(\boldsymbol{\theta} - \boldsymbol{\theta}_k) \tag{21}$$

We then take the gradient with respect to $\boldsymbol{\theta}$ for the RHS of the Eq. (21), we can obtain the following:

$$\nabla_{\boldsymbol{\theta}}\mathcal{L}_{CL}(\boldsymbol{\theta}_k) - \alpha F(\boldsymbol{\theta} - \boldsymbol{\theta}_k) = 0 \tag{22}$$

Solving the above equation leads to the following unlearning for the previously learned tasks:

$$\boldsymbol{\theta}_k' = \boldsymbol{\theta}_k + \frac{1}{\alpha}F^{-1}\nabla_{\boldsymbol{\theta}}\mathcal{L}_{CL}(\boldsymbol{\theta}_k) \tag{23}$$

Equation (23) is nearly identical to Equation (17), with the only distinction being that Equation (17) incorporates an additional random noise perturbation, which helps the CL model escape local minima Raginsky et al. (2017) and saddle point Ge et al. (2015). The constant $\frac{1}{\alpha}$ now takes on a new interpretation, serving as the unlearning rate.

In summary, we name our proposed method as *refresh*, which reflects our new learning mechanism that avoids learning outdated information. Algorithm 1 presents the general refresh learning method with a unlearn-relearn framework for general CL. Line 3-5 describes the unlearn step for current loss at each CL step. Line 6 describes the relearn step for current loss.

# 4 THEORETICAL ANALYSIS

Our method can be interpreted theoretically and improves the generalization of CL by improving the flatness of the loss landscape. Specifically, *refresh learning* can be characterized as the FIM weighted gradient norm penalized optimization by the following theorem.

**Theorem 4.1.** *With one step of unlearning by Eq. (17), refresh learning approximately minimize the following FIM weighted gradient norm of the loss function. That is, solving Eq. (11) and Eq. (12) approximately solves the following optimization:*

$$\min_{\boldsymbol{\theta}} \mathcal{L}^{CL}(\boldsymbol{\theta}) + \sigma||\nabla\mathcal{L}^{CL}(\boldsymbol{\theta})F^{-1}|| \qquad (24)$$

*where $\sigma > 0$ is a constant.*

The above theorem shows that *refresh learning* seeks to minimize the FIM weighted gradient norm of the loss function. This optimization objective promotes the flatness of the loss landscape since a smaller FIM weighted gradient norm indicates flatter loss landscape. In practice, flatter loss landscape has been demonstrated with significantly improved generalization (Izmailov et al., 2018). It is important to note that our method is more flexible and efficient than minimizing the FIM weighted gradient norm of the loss function since we can flexibly control the degree of unlearning with different number of steps, which may involve higher order flatness of loss landscape. Furthermore, optimizing Eq. (24) necessitates the calculation of the Hessian matrix, a computationally intensive task. In contrast, our method offers a significant efficiency advantage as it does not require the computation of the Hessian matrix. Due to the space limitations, we put detailed theorem proof in Appendix B.

# 5 EXPERIMENTS

## 5.1 SETUP

**Datasets** We perform experiments on various datasets, including CIFAR10 (10 classes), CIFAR100 (100 classes), Tiny-ImageNet (200 classes) and evaluate the effectiveness of our proposed methods in task incremental learning (Task-IL) and class incremental learning (Class-IL). Following Buzzega et al. (2020), we divided the CIFAR-10 dataset into five separate tasks, each containing two distinct classes. Similarly, we partitioned the CIFAR-100 dataset into ten tasks, each has ten classes. Additionally, for Tiny-ImageNet, we organized it into ten tasks, each has twenty classes.

**Baselines** We compare to the following baseline methods for comparisons. (1) Regularization-based methods, including oEWC (Schwarz et al., 2018), synaptic intelligence (SI) (Zenke et al., 2017a), Learning without Forgetting (LwF) (Li & Hoiem, 2018), Classifier-Projection Regularization (CPR) (Cha et al., 2021), Gradient Projection Memory (GPM) (Saha et al., 2021). (2) Bayesian-based methods, NCL (Kao et al., 2021). (3) Architecture-based methods, including HAT (Serra et al., 2018). (4) Memory-based methods, including ER (Chaudhry et al., 2019b), A-GEM (Chaudhry et al., 2019a), GSS (Aljundi et al., 2019), DER++ (Buzzega et al., 2020), HAL(Chaudhry et al., 2021).

**Implementation Details** We use ResNet18 (He et al., 2016) on the above datasets. We adopt the hyperparameters from the DER++ codebase (Buzzega et al., 2020) as the baseline settings for all the methods we compared in the experiments. Additionally, to enhance runtime efficiency in our approach, we implemented the refresh mechanism, which runs every two iterations.

**Evaluation Metrics** We evaluate the performance of proposed *refresh* method by integrating with several existing methods with (1) overall accuracy (ACC), which is the average accuracy across the entire task sequence and (2) backward transfer (BWT), which measures the amount of forgetting on previously learned tasks. If BWT > 0, which means learning on current new task is helpful for improving the performance of previously learned tasks. If BWT ≤ 0, which means learning on current new task can lead to forgetting previously learned tasks. Each experiment result is averaged for 10 runs with mean and standard deviation.

## 5.2 RESULTS

We present the overall accuracy for task-IL and class-IL in Table 2. Due to space limitations, we put BWT results in Table 9 in Appendix C.5. We can observe that with the refresh plug-in, the

Table 2: **Task-IL and class-IL** overall accuracy on CIFAR10, CIFAR-100 and Tiny-ImageNet, respectively with memory size 500. '—' indicates not applicable.

| Algorithm Method | CIFAR-10 | | CIFAR-100 | | Tiny-ImageNet | |
|---|---|---|---|---|---|---|
| | Class-IL | Task-IL | Class-IL | Task-IL | Class-IL | Task-IL |
| fine-tuning | $19.62 \pm 0.05$ | $61.02 \pm 3.33$ | $9.29 \pm 0.33$ | $33.78 \pm 0.42$ | $7.92 \pm 0.26$ | $18.31 \pm 0.68$ |
| Joint train | $92.20 \pm 0.15$ | $98.31 \pm 0.12$ | $71.32 \pm 0.21$ | $91.31 \pm 0.17$ | $59.99 \pm 0.19$ | $82.04 \pm 0.10$ |
| SI | $19.48 \pm 0.17$ | $68.05 \pm 5.91$ | $9.41 \pm 0.24$ | $31.08 \pm 1.65$ | $6.58 \pm 0.31$ | $36.32 \pm 0.13$ |
| LwF | $19.61 \pm 0.05$ | $63.29 \pm 2.35$ | $9.70 \pm 0.23$ | $28.07 \pm 1.96$ | $8.46 \pm 0.22$ | $15.85 \pm 0.58$ |
| NCL | $19.53 \pm 0.32$ | $64.49 \pm 4.06$ | $8.12 \pm 0.28$ | $20.92 \pm 2.32$ | $7.56 \pm 0.36$ | $16.29 \pm 0.87$ |
| GPM | —— | $90.68 \pm 3.29$ | —— | $72.48 \pm 0.40$ | —— | —— |
| UCB | —— | $79.28 \pm 1.87$ | —— | $57.15 \pm 1.67$ | —— | —— |
| HAT | —— | $92.56 \pm 0.78$ | —— | $72.06 \pm 0.50$ | —— | —— |
| A-GEM | $22.67 \pm 0.57$ | $89.48 \pm 1.45$ | $9.30 \pm 0.32$ | $48.06 \pm 0.57$ | $8.06 \pm 0.04$ | $25.33 \pm 0.49$ |
| GSS | $49.73 \pm 4.78$ | $91.02 \pm 1.57$ | $13.60 \pm 2.98$ | $57.50 \pm 1.93$ | —— | —— |
| HAL | $41.79 \pm 4.46$ | $84.54 \pm 2.36$ | $9.05 \pm 2.76$ | $42.94 \pm 1.80$ | —— | —— |
| oEWC | $19.49 \pm 0.12$ | $64.31 \pm 4.31$ | $8.24 \pm 0.21$ | $21.2 \pm 2.08$ | $7.42 \pm 0.31$ | $15.19 \pm 0.82$ |
| oEWC+refresh | $\mathbf{20.37 \pm 0.65}$ | $\mathbf{66.89 \pm 2.57}$ | $\mathbf{8.78 \pm 0.42}$ | $\mathbf{23.31 \pm 1.87}$ | $\mathbf{7.83 \pm 0.15}$ | $\mathbf{17.32 \pm 0.85}$ |
| CPR(EWC) | $19.61 \pm 3.67$ | $65.23 \pm 3.87$ | $8.42 \pm 0.37$ | $21.43 \pm 2.57$ | $7.67 \pm 0.23$ | $15.58 \pm 0.91$ |
| CPR(EWC)+refresh | $\mathbf{20.53 \pm 2.42}$ | $\mathbf{67.36 \pm 3.68}$ | $\mathbf{9.06 \pm 0.58}$ | $\mathbf{22.90 \pm 1.71}$ | $\mathbf{8.06 \pm 0.43}$ | $\mathbf{17.90 \pm 0.77}$ |
| ER | $57.74 \pm 0.27$ | $93.61 \pm 0.27$ | $20.98 \pm 0.35$ | $73.37 \pm 0.43$ | $9.99 \pm 0.29$ | $48.64 \pm 0.46$ |
| ER+refresh | $\mathbf{61.86 \pm 1.35}$ | $\mathbf{94.15 \pm 0.46}$ | $\mathbf{22.23 \pm 0.73}$ | $\mathbf{75.45 \pm 0.67}$ | $\mathbf{11.09 \pm 0.46}$ | $\mathbf{50.85 \pm 0.53}$ |
| DER++ | $72.70 \pm 1.36$ | $93.88 \pm 0.50$ | $36.37 \pm 0.85$ | $75.64 \pm 0.60$ | $19.38 \pm 1.41$ | $51.91 \pm 0.68$ |
| DER+++refresh | $\mathbf{74.42 \pm 0.82}$ | $\mathbf{94.64 \pm 0.38}$ | $\mathbf{38.49 \pm 0.76}$ | $\mathbf{77.71 \pm 0.85}$ | $\mathbf{20.81 \pm 1.28}$ | $\mathbf{54.06 \pm 0.79}$ |

performance of all compared methods can be further significantly improved. Notably, compared to the strong baseline DER++, our method improves by more than 2% in many cases on CIFAR10, CIFAR100 and Tiny-ImageNet. The performance improvement demonstrates the effectiveness and general applicability of refresh mechanism, which can more effectively retain important information from previously learned tasks since it can more effectively utilize model capacity to perform CL.

## 5.3 ABLATION STUDY AND HYPERPARAMETER ANALYSIS

**Hyperparameter Analysis** We evaluate the sensitivity analysis of the hyperparameters, the unlearning rate $\gamma$ and the number of unlearning steps $J$ in Table 5 in Appendix. We can observe that with increasing number of unlearning steps $J$, the CL performance only slightly improves and then decreases but with higher computation cost. For computation efficiency, we only choose one step of unlearning. We also evaluate the effect of the unlearning rate $\gamma$ to the CL model performance.

**Effect of Memory Size** To evaluate the effect of different memory buffer size, we provide results in Table 4 in Appendix. The results show that with larger memory size of 2000, our refresh plug-in also substantially improves the compared methods.

**Computation Efficiency** To evaluate the efficiency of the proposed method, we evaluate and compare DER+++refresh learning with DER++ on CIFAR100 in Table 8 in Appendix. This running time indicates that *refresh learning* increases $0.81\times$ cost compared to the baseline without refresh learning. This shows our method is efficient and only introduces marginal computation cost.

## 6 CONCLUSION

This paper introduces an unified framework for CL. and unifies various existing CL approaches as special cases. Additionally, the paper introduces a novel approach called *refresh learning*, which draws inspiration from neuroscience principles and seamlessly integrates with existing CL methods, resulting in enhanced generalization performance. The effectiveness of the proposed framework and the novel refresh learning method is substantiated through a series of extensive experiments on various CL datasets. This research represents a significant advancement in CL, offering a unified and adaptable solution.

**Acknowledgments** This work was partially supported by NSF IIS 2347592, 2347604, 2348159, 2348169, DBI 2405416, CCF 2348306, CNS 2347617.

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

## Appendix

## A    RECAST EXISTING CL METHODS INTO OUR UNIFIED AND GENERAL FRAMEWORK

$$\mathcal{L}^{CL} = \mathcal{L}_{CE}(\boldsymbol{x}, y) + \alpha D_{\boldsymbol{\Phi}}(h_{\boldsymbol{\theta}}(\boldsymbol{x}), \boldsymbol{z}) + \beta D_{\boldsymbol{\Psi}}(\boldsymbol{\theta}, \boldsymbol{\theta}_{old}) \tag{25}$$

The following is the definition of Bregman divergence:

$$D_{\boldsymbol{\Phi}}(\boldsymbol{p}, \boldsymbol{q}) = \boldsymbol{\Phi}(\boldsymbol{p}) - \boldsymbol{\Phi}(\boldsymbol{q}) - \langle \nabla \boldsymbol{\Phi}(\boldsymbol{q}), \boldsymbol{p} - \boldsymbol{q} \rangle \tag{26}$$

### A.1    CAST CPR INTO THE GENERAL FRAMEWORK

In Eq. (25), we take $\boldsymbol{\Phi}(\boldsymbol{p}) = \sum_{i=1}^{i=n} \boldsymbol{p}_i \log \boldsymbol{p}_i$. Here, $\boldsymbol{p}$ and $\boldsymbol{q}$ are probability simplex, i.e., $\sum_{i=1}^{i=n} \boldsymbol{p}_i = 1$ and $\sum_{i=1}^{i=n} \boldsymbol{q}_i = 1$. Then, we plug $\boldsymbol{\Phi}(\boldsymbol{p})$ into Eq. (26). We can obtain the following equation:

$$D_{\boldsymbol{\Phi}}(\boldsymbol{p}, \boldsymbol{q}) = \sum_{i=1}^{i=n} \boldsymbol{p}_i \log \boldsymbol{p}_i - \sum_{i=1}^{i=n} \boldsymbol{q}_i \log \boldsymbol{q}_i - \langle \log(\boldsymbol{q}) + 1, \boldsymbol{p} - \boldsymbol{q} \rangle \tag{27}$$

$$= \sum_{i=1}^{i=n} \boldsymbol{p}_i \log \boldsymbol{p}_i - \sum_{i=1}^{i=n} \boldsymbol{p}_i \log \boldsymbol{q}_i - \sum_{i=1}^{i=n} \boldsymbol{p}_i + \sum_{i=1}^{i=n} \boldsymbol{q}_i \tag{28}$$

$$= \sum_{i=1}^{i=n} \boldsymbol{p}_i \log \frac{\boldsymbol{p}_i}{\boldsymbol{q}_i} \tag{29}$$

$$= -H(\boldsymbol{p}) + H(\boldsymbol{p}, \boldsymbol{q}) \tag{30}$$

$$= \mathbb{KL}(\boldsymbol{p} || \boldsymbol{q}) \tag{31}$$

where $H(\boldsymbol{p})$ is the entropy for the probability distribution $\boldsymbol{p}$. and $H(\boldsymbol{p}, \boldsymbol{q})$ is the cross entropy between probability distributions $\boldsymbol{p}$ and $\boldsymbol{q}$.

When we take the probability distribution $\boldsymbol{p} = g_{\boldsymbol{\theta}}(\boldsymbol{x})$ , i.e., the CL model output probability distribution over the classes, and $\boldsymbol{q} = \boldsymbol{v}$, i.e., the uniform distribution over the underlying classes, $D_{\boldsymbol{\Phi}}(\boldsymbol{p}, \boldsymbol{q}) = \mathbb{KL}(g_{\boldsymbol{\theta}}(\boldsymbol{x}), \boldsymbol{v})$. This precisely recovers the CPR method.

### A.2    CAST EWC INTO THE GENERAL FRAMEWORK

In Eq. (25), we set $\alpha = 0$, we take $\boldsymbol{\Psi}(\boldsymbol{\theta}) = \frac{1}{2} \boldsymbol{\theta}^T F \boldsymbol{\theta}$. We set $\boldsymbol{p} = \boldsymbol{\theta}$ and $\boldsymbol{q} = \boldsymbol{\theta}_{old}$. where $F$ is the diagonal Fisher information matrix.

$$D_{\boldsymbol{\Phi}}(\boldsymbol{p}, \boldsymbol{q}) = \boldsymbol{\Phi}(\boldsymbol{p}) - \boldsymbol{\Phi}(\boldsymbol{q}) - \langle \nabla \boldsymbol{\Phi}(\boldsymbol{q}), \boldsymbol{p} - \boldsymbol{q} \rangle \tag{32}$$

$$= \frac{1}{2} \boldsymbol{\theta}^T F \boldsymbol{\theta} - \frac{1}{2} \boldsymbol{\theta}_{old}^T F \boldsymbol{\theta}_{old} - \langle \boldsymbol{\theta}_{old} F, \boldsymbol{\theta} - \boldsymbol{\theta}_{old} \rangle \tag{33}$$

$$= \frac{1}{2} \boldsymbol{\theta}^T F \boldsymbol{\theta} + \frac{1}{2} \boldsymbol{\theta}_{old}^T F \boldsymbol{\theta}_{old} - \langle \boldsymbol{\theta}_{old} F, \boldsymbol{\theta} \rangle \tag{34}$$

$$= \frac{1}{2} (\boldsymbol{\theta} - \boldsymbol{\theta}_{old})^T F (\boldsymbol{\theta} - \boldsymbol{\theta}_{old}) \tag{35}$$

Then, we recover the EWC method.

### A.3    CAST DER INTO THE GENERAL FRAMEWORK

In Eq. (25), we set $\beta = 0$ and take $\boldsymbol{\Phi}(\boldsymbol{p}) = ||\boldsymbol{p}||^2$

$$D_{\boldsymbol{\Phi}}(\boldsymbol{p}, \boldsymbol{q}) = \boldsymbol{\Phi}(\boldsymbol{p}) - \boldsymbol{\Phi}(\boldsymbol{q}) - \langle \nabla \boldsymbol{\Phi}(\boldsymbol{q}), \boldsymbol{p} - \boldsymbol{q} \rangle \tag{36}$$

$$= ||\boldsymbol{p}||^2 - ||\boldsymbol{q}||^2 - \langle 2\boldsymbol{q}, \boldsymbol{p} - \boldsymbol{q} \rangle \tag{37}$$

$$= ||\boldsymbol{p}||^2 + ||\boldsymbol{q}||^2 - 2\langle \boldsymbol{p}, \boldsymbol{q} \rangle \tag{38}$$

$$= (\boldsymbol{p} - \boldsymbol{q})^2 \tag{39}$$

Next, we take $\boldsymbol{p} = u_{\boldsymbol{\theta}}(\boldsymbol{x})$ and $\boldsymbol{q} = \boldsymbol{z}$. $D_{\boldsymbol{\Phi}}(\boldsymbol{p}, \boldsymbol{q}) = ||u_{\boldsymbol{\theta}}(\boldsymbol{x}) - \boldsymbol{z}||^2$. Then, we recover the DER method.

### A.4 CAST ER INTO THE GENERAL FRAMEWORK

In Eq. (25), we set $\beta = 0$, we take $\boldsymbol{p} = \boldsymbol{y}$, i.e., the one-hot representation of the ground-truth label and $\boldsymbol{q} = g_{\boldsymbol{\theta}}(\boldsymbol{x})$, i.e., the CL model output probability distribution over the classes. Then, $D_{\boldsymbol{\Phi}}(\boldsymbol{p}, \boldsymbol{q})$ is equivalent to the cross-entropy loss $H(\boldsymbol{p}, \boldsymbol{q})$ according to Eq. (30). As a result, we recover the ER method.

### A.5 CAST VCL INTO THE GENERAL FRAMEWORK

In Eq. (25), we set $\alpha = 0$, we take $\boldsymbol{\Psi}(p) = \int p(\boldsymbol{\theta}) \log p(\boldsymbol{\theta}) d\boldsymbol{\theta}$. $\int p(\boldsymbol{\theta}) d\boldsymbol{\theta} = 1$. Then, the following Bregman divergence can be expressed as:

$$D_{\boldsymbol{\Psi}}(p, q) = \boldsymbol{\Psi}(p) - \boldsymbol{\Psi}(q) - \langle \nabla \boldsymbol{\Psi}(q), p - q \rangle \tag{40}$$

$$= \int p(\boldsymbol{\theta}) \log p(\boldsymbol{\theta}) d\boldsymbol{\theta} - \int q(\boldsymbol{\theta}) \log q(\boldsymbol{\theta}) d\boldsymbol{\theta} - \int (1 + \log q(\boldsymbol{\theta}))(p(\boldsymbol{\theta}) - q(\boldsymbol{\theta})) d\boldsymbol{\theta} \tag{41}$$

$$= \int p(\boldsymbol{\theta}) \log p(\boldsymbol{\theta}) d\boldsymbol{\theta} - \int p(\boldsymbol{\theta}) \log q(\boldsymbol{\theta}) d\boldsymbol{\theta} \tag{42}$$

$$= \int p(\boldsymbol{\theta}) \log \frac{p(\boldsymbol{\theta})}{q(\boldsymbol{\theta})} d\boldsymbol{\theta} \tag{43}$$

$$= \mathbb{KL}(p(\boldsymbol{\theta}) || q(\boldsymbol{\theta})) \tag{44}$$

Variational continual learning (VCL) Nguyen et al. (2018) is a Bayesian-based method for mitigating forgetting in CL. The basic idea of VCL is to constrain the current model parameter distribution to be close to that of previous tasks. It optimizes the following objective.

$$\mathcal{L}^{CL} = \mathcal{L}_{CE}(\boldsymbol{x}, y) + \beta \mathbb{KL}(P(\boldsymbol{\theta}|\mathcal{D}_{1:t}), P(\boldsymbol{\theta}_{old}|\mathcal{D}_{1:t-1})) \tag{45}$$

where $\mathcal{D}_{1:t}$ denotes the dataset from task 1 to task $t$. $P(\boldsymbol{\theta}|\mathcal{D}_{1:t})$ is the posterior distribution of the model parameters on the entire task sequence $\mathcal{D}_{1:t}$. $P(\boldsymbol{\theta}_{old}|\mathcal{D}_{1:t-1})$ is the posterior distribution of the model parameters on the tasks $\mathcal{D}_{1:t-1}$. In this case, $P(\boldsymbol{\theta}|\mathcal{D}_{1:t})$ and $P(\boldsymbol{\theta}_{old}|\mathcal{D}_{1:t-1})$ are both continuous distributions. In this case, in Eq. (2), we set $\alpha = 0$. we take $\boldsymbol{\Psi}$ to be $\boldsymbol{\Psi}(p) = \int p(\boldsymbol{\theta}) \log p(\boldsymbol{\theta}) d\boldsymbol{\theta}$. We then set $p = P(\boldsymbol{\theta}|\mathcal{D}_{1:t})$ and $q = P(\boldsymbol{\theta}_{old}|\mathcal{D}_{1:t-1})$. We then recover the VCL method.

### A.6 CAST NATURAL GRADIENT CL INTO THE GENERAL FRAMEWORK

In Eq. (25), we adopt the first-order Taylor expansion on the first loss term as the following:

$$\mathcal{L}_{CE}(\boldsymbol{\theta}) \approx \mathcal{L}_{CE}(\boldsymbol{\theta}_k) + \nabla_{\boldsymbol{\theta}} \mathcal{L}_{CE}(\boldsymbol{\theta}_k)(\boldsymbol{\theta} - \boldsymbol{\theta}_k) \tag{46}$$

For the second loss term in Eq. (25), we take $\boldsymbol{\Phi}(\boldsymbol{p}) = \sum_{i=1}^{i=n} \boldsymbol{p}_i \log \boldsymbol{p}_i$. $\boldsymbol{z}$ to be the ground truth one-hot vector for the labeled data. Then, the second loss term is the cross entropy loss on the

previously learned tasks. We adopt the second-order Taylor expansion on the second loss term as the following:

$$D_{\Phi}(h_{\theta}(x), z) \approx D_{\Phi}(h_{\theta_k}(x), z) + \nabla_{\theta} D_{\Phi}(h_{\theta_k}(x), z)(\theta - \theta_k) + \frac{1}{2}(\theta - \theta_k)^T F(\theta - \theta_k) \quad (47)$$

where $F$ is the Fisher information matrix (FIM) of the loss $D_{\Phi}(h_{\theta}(x), z)$ on previously learned tasks. Since $\nabla_{\theta} D_{\Phi}(h_{\theta}(x), z)$ is close to zero at the stationary point, i.e., $\theta_k$, we thus only need to optimize the quadratic term in Eq. 47.

For the third loss term in Eq. (25), we adopt the $\Psi = \frac{1}{2}||\theta||^2$. Thus, the third loss term becomes $D_{\Psi}(\theta, \theta_k) = \frac{1}{2}||\theta - \theta_k||^2$.

In summary, the approximate loss function for Eq. (25) can be expressed as the following:

$$\nabla_{\theta} \mathcal{L}_{CE}(\theta_k)(\theta - \theta_k) + \frac{\alpha}{2}(\theta - \theta_k)^T F(\theta - \theta_k) + \frac{\beta}{2}||\theta - \theta_k||^2 \quad (48)$$

We then apply first-order gradient method on the Eq. (48), we can obtain the following:

$$\nabla_{\theta} \mathcal{L}_{CE}(\theta_k) + \alpha(\theta - \theta_k)F + \beta(\theta - \theta_k) = 0 \quad (49)$$

We can obtain the following.

$$(\alpha F + \beta I)\theta = (\alpha F + \beta I)(\theta_k - ((\alpha F + \beta I)^{-1}\nabla_{\theta}\mathcal{L}_{CE}(\theta_k))) \quad (50)$$

We can get the following natural gradient CL method.

$$\theta_{k+1} = \theta_k - ((\alpha F + \beta I)^{-1}\nabla_{\theta}\mathcal{L}_{CE}(\theta_k)) \quad (51)$$

when $\beta = 0$, the above equation recover the standard natural gradient CL method without damping as the following:

$$\theta_{k+1} = \theta_k - (\alpha F)^{-1}\nabla_{\theta}\mathcal{L}_{CE}(\theta_k) \quad (52)$$

## B  THEOREM PROOF

*Proof. Proof sketch:* We outline our proof in the following. We denote the weighted gradient norm regularized CL loss function as $\mathcal{L}_{GN}(\theta)$ and our refresh learning loss function as $\mathcal{L}_{refresh}(\theta)$. Then, we calculate their gradient $\nabla_{\theta}\mathcal{L}_{GN}(\theta)$ and $\nabla_{\theta}\mathcal{L}_{refresh}(\theta)$, respectively. Finally, we show their gradient is approximately the same, i.e., $\nabla_{\theta}\mathcal{L}_{GN}(\theta) \approx \nabla_{\theta}\mathcal{L}_{refresh}(\theta)$, then the conclusion follows.

**(1) Calculate the gradient $\nabla_{\theta}\mathcal{L}_{GN}(\theta)$**   We define the gradient norm regularized CL loss function as:

$$\mathcal{L}_{GN}(\theta) = \mathcal{L}^{CL}(\theta) + \sigma||\nabla_{\theta}\mathcal{L}^{CL}(\theta)F^{-1}|| \quad (53)$$

Then, we take the derivative with respect to $\theta$ in Eq. (53), we got the following:

$$\nabla_{\theta}||\nabla_{\theta}\mathcal{L}^{CL}(\theta)F^{-1}|| = \nabla_{\theta}(||\nabla_{\theta}\mathcal{L}^{CL}(\theta)F^{-1}||^2)^{\frac{1}{2}} \quad (54)$$

$$= \frac{1}{2}(||\nabla_{\theta}\mathcal{L}^{CL}(\theta)F^{-1}||^2)^{-\frac{1}{2}}(2\nabla_{\theta}\mathcal{L}^{CL}(\theta)F^{-1})\nabla_{\theta}^2\mathcal{L}^{CL}(\theta)F^{-1} \quad (55)$$

$$= \frac{\nabla_{\theta}\mathcal{L}^{CL}(\theta)F^{-1}\nabla_{\theta}^2\mathcal{L}^{CL}(\theta)F^{-1}}{||\nabla_{\theta}\mathcal{L}^{CL}(\theta)F^{-1}||} \quad (56)$$

$$\approx \nabla_{\theta}^2\mathcal{L}^{CL}(\theta)F^{-1}\frac{\nabla_{\theta}\mathcal{L}^{CL}(\theta)}{||\nabla_{\theta}\mathcal{L}^{CL}(\theta)||} \quad (57)$$

Table 4: **Task-IL and class-IL** overall accuracy on CIFAR-100 and Tiny-ImageNet, respectively with memory size 2000. '—' indicates not applicable.

| Algorithm | CIFAR-100 | | Tiny-ImageNet | |
|---|---|---|---|---|
| Method | Class-IL | Task-IL | Class-IL | Task-IL |
| ER | $36.06 \pm 0.72$ | $81.09 \pm 0.45$ | $15.16 \pm 0.78$ | $58.19 \pm 0.69$ |
| ER+refresh | $\mathbf{37.29 \pm 0.85}$ | $\mathbf{83.21 \pm 1.23}$ | $\mathbf{16.93 \pm 0.86}$ | $\mathbf{59.42 \pm 0.51}$ |
| DER++ | $50.72 \pm 0.71$ | $82.43 \pm 0.38$ | $24.21 \pm 1.09$ | $62.22 \pm 0.87$ |
| DER+++refresh | $\mathbf{52.81 \pm 0.80}$ | $\mathbf{84.05 \pm 0.77}$ | $\mathbf{27.37 \pm 1.53}$ | $\mathbf{64.31 \pm 0.98}$ |

**(2) Calculate the gradient $\nabla_{\boldsymbol{\theta}} \mathcal{L}_{refresh}(\boldsymbol{\theta})$** Then, we define the *refresh learning* loss function as the following:

$$\mathcal{L}_{refresh} = \mathcal{L}^{CL}(\boldsymbol{\theta} + s\boldsymbol{\delta}) \tag{58}$$

where, we set $\boldsymbol{\delta} = F^{-1} \frac{\nabla_{\boldsymbol{\theta}} \mathcal{L}^{CL}(\boldsymbol{\theta})}{||\nabla_{\boldsymbol{\theta}} \mathcal{L}^{CL}(\boldsymbol{\theta})||} + \mathcal{N}(0, 2\gamma F^{-1})$. Then, we take the first-order Taylor expansion on $\mathcal{L}_{refresh}$ Zhao et al. (2022) as the following:

$$\nabla_{\boldsymbol{\theta}} \mathcal{L}^{CL}(\boldsymbol{\theta} + s\boldsymbol{\delta}) \approx \nabla_{\boldsymbol{\theta}} \mathcal{L}^{CL}(\boldsymbol{\theta}) + \nabla_{\boldsymbol{\theta}}^2 \mathcal{L}^{CL}(\boldsymbol{\theta}) s\boldsymbol{\delta} \tag{59}$$

$$\nabla_{\boldsymbol{\theta}}^2 \mathcal{L}^{CL}(\boldsymbol{\theta})\boldsymbol{\delta} = \nabla_{\boldsymbol{\theta}}^2 \mathcal{L}^{CL}(\boldsymbol{\theta}) F^{-1} \frac{\nabla_{\boldsymbol{\theta}} \mathcal{L}^{CL}(\boldsymbol{\theta})}{||\nabla_{\boldsymbol{\theta}} \mathcal{L}^{CL}(\boldsymbol{\theta})||} + \mathcal{N}(0, 2\gamma[\nabla_{\boldsymbol{\theta}}^2 \mathcal{L}^{CL}(\boldsymbol{\theta})]^2 F^{-1}) \tag{60}$$

**(3) Show that these two loss gradients are approximately the same, i.e., $\nabla_{\boldsymbol{\theta}} \mathcal{L}_{GN}(\boldsymbol{\theta}) \approx \nabla_{\boldsymbol{\theta}} \mathcal{L}_{refresh}(\boldsymbol{\theta})$**

$$\nabla_{\boldsymbol{\theta}} \mathcal{L}^{CL}(\boldsymbol{\theta} + s\boldsymbol{\delta}) \approx \nabla_{\boldsymbol{\theta}} \mathcal{L}^{CL}(\boldsymbol{\theta}) + \nabla_{\boldsymbol{\theta}}^2 \mathcal{L}^{CL}(\boldsymbol{\theta}) s\boldsymbol{\delta} \tag{61}$$

$$= \nabla_{\boldsymbol{\theta}} \mathcal{L}^{CL}(\boldsymbol{\theta}) + s\nabla_{\boldsymbol{\theta}}^2 \mathcal{L}^{CL}(\boldsymbol{\theta}) F^{-1} \frac{\nabla_{\boldsymbol{\theta}} \mathcal{L}^{CL}(\boldsymbol{\theta})}{||\nabla_{\boldsymbol{\theta}} \mathcal{L}^{CL}(\boldsymbol{\theta})||} + \mathcal{N}(0, 2\gamma s^2 [\nabla_{\boldsymbol{\theta}}^2 \mathcal{L}^{CL}(\boldsymbol{\theta})]^2 F^{-1}) \tag{62}$$

$$\approx \nabla_{\boldsymbol{\theta}} \mathcal{L}^{CL}(\boldsymbol{\theta}) + s\nabla_{\boldsymbol{\theta}} ||\nabla_{\boldsymbol{\theta}} \mathcal{L}^{CL}(\boldsymbol{\theta}) F^{-1}|| + \mathcal{N}(0, 2\gamma[\nabla_{\boldsymbol{\theta}}^2 \mathcal{L}^{CL}(\boldsymbol{\theta})]^2 F^{-1}) \tag{63}$$

where the additional random Gaussian noise in Eq. (63) helps the CL model escape local minima and saddle points to achieve global minima solution. Furthermore, Eq. (63) indicates that

$$\nabla_{\boldsymbol{\theta}} \mathcal{L}_{refresh} \approx \nabla_{\boldsymbol{\theta}} [\mathcal{L}^{CL}(\boldsymbol{\theta}) + \sigma ||\nabla_{\boldsymbol{\theta}} \mathcal{L}^{CL}(\boldsymbol{\theta}) F^{-1}||] = \nabla_{\boldsymbol{\theta}} \mathcal{L}_{GN}(\boldsymbol{\theta}) \tag{64}$$

In other words, the gradient of the refresh learning approximately the same as that of weighted gradient norm regularized CL loss function. The conclusion then follows. □

## C MORE EXPERIMENTAL RESULTS

### C.1 RESULTS ON MNIST

Table 3: **Domain-IL** overall accuracy on **P-MNIST** and **R-MNIST**, respectively with memory size 500.

| Algorithm | P-MNIST | R-MNIST |
|---|---|---|
| Method | Domain-IL | Domain-IL |
| oEWC | $59.57 \pm 2.37$ | $77.35 \pm 5.77$ |
| oEWC+refresh | $\mathbf{61.23 \pm 2.18}$ | $\mathbf{79.21 \pm 4.98}$ |
| ER | $78.45 \pm 0.72$ | $88.91 \pm 1.44$ |
| ER+refresh | $\mathbf{80.28 \pm 1.06}$ | $\mathbf{90.53 \pm 1.67}$ |
| DER++ | $88.21 \pm 0.39$ | $92.77 \pm 1.05$ |
| DER+++refresh | $\mathbf{88.93 \pm 0.58}$ | $\mathbf{93.28 \pm 0.75}$ |

## C.2 RESULTS WITH MEMORY SIZE OF 2000

## C.3 HYPERPARAMETER ANALYSIS

Table 5: Analysis of unlearning rate $\gamma$ and number of unlearning steps $J$ on CIFAR100 with task-IL.

| $\gamma$ | 0.02 | 0.03 | 0.04 |
|---|---|---|---|
| Accuracy | $77.23 \pm 0.97$ | $77.71 \pm 0.85$ | $77.08 \pm 0.90$ |
| $J$ | 1 | 2 | 3 |
| Accuracy | $77.71 \pm 0.85$ | $77.76 \pm 0.82$ | $75.93 \pm 1.06$ |

Table 6: Analysis of unlearning rate $\gamma$ and number of unlearning steps $J$ on **CIFAR10** with task-IL.

| $\gamma$ | 0.02 | 0.03 | 0.04 |
|---|---|---|---|
| Accuracy | $94.27 \pm 0.42$ | $94.64 \pm 0.38$ | $94.82 \pm 0.51$ |
| $J$ | 1 | 2 | 3 |
| Accuracy | $94.64 \pm 0.38$ | $94.73 \pm 0.43$ | $93.50 \pm 0.57$ |

Table 7: Analysis of unlearning rate $\gamma$ and number of unlearning steps $J$ on **Tiny-ImageNet** with task-IL.

| $\gamma$ | 0.02 | 0.03 | 0.04 |
|---|---|---|---|
| Accuracy | $53.27 \pm 0.72$ | $54.06 \pm 0.79$ | $54.21 \pm 0.83$ |
| $J$ | 1 | 2 | 3 |
| Accuracy | $54.06 \pm 0.79$ | $54.17 \pm 0.91$ | $52.29 \pm 0.86$ |

## C.4 COMPUTATION EFFICIENCY

Table 8: Computational efficiency of *refresh learning* on CIFAR100 with one epoch training

| CIFAR100 | DER++ | DER+++refresh |
|---|---|---|
| running time (seconds) | 8.4 | 15.2 |

## C.5 BACKWARD TRANSFER

We evaluate Backward Transfer (BWT) in Table 9.

Table 9: **Backward Transfer** of various methods with memory size 500.

| Method | CIFAR10 | | CIFAR100 | | Tiny-ImageNet | |
|---|---|---|---|---|---|---|
| | Class-IL | Task-IL | Class-IL | Task-IL | Class-IL | Task-IL |
| finetuning | $-96.39 \pm 0.12$ | $-46.24 \pm 2.12$ | $-89.68 \pm 0.96$ | $-62.46 \pm 0.78$ | $-78.94 \pm 0.81$ | $-67.34 \pm 0.79$ |
| AGEM | $-94.01 \pm 1.16$ | $-14.26 \pm 1.18$ | $-88.5 \pm 1.56$ | $-45.43 \pm 2.32$ | $-78.03 \pm 0.78$ | $-59.28 \pm 1.08$ |
| GSS | $-62.88 \pm 2.67$ | $-7.73 \pm 3.99$ | $-82.17 \pm 4.16$ | $-33.98 \pm 1.54$ | —— | —— |
| HAL | $-62.21 \pm 4.34$ | $-5.41 \pm 1.10$ | $-49.29 \pm 2.82$ | $-13.60 \pm 1.04$ | —— | —— |
| ER | $-45.35 \pm 0.07$ | **-3.54 ± 0.35** | $-74.84 \pm 1.38$ | $-16.81 \pm 0.97$ | $-75.24 \pm 0.76$ | $-31.98 \pm 1.35$ |
| ER+refresh | **-40.89 ± 0.86** | $-3.97 \pm 0.38$ | **-73.78 ± 1.59** | **-15.65 ± 0.87** | **-74.49 ± 0.80** | **-30.06 ± 1.51** |
| DER++ | $-22.38 \pm 4.41$ | $-4.66 \pm 1.15$ | $-53.89 \pm 1.85$ | $-14.72 \pm 0.96$ | $-64.6 \pm 0.56$ | $-27.21 \pm 1.23$ |
| DER++ refresh | **-22.03 ± 3.89** | **-4.37 ± 1.25** | **-53.51 ± 0.70** | **-14.23 ± 0.75** | **-63.90 ± 0.61** | **-25.05 ± 1.05** |

