# OpenReview forum: "A Unified and General Framework for Continual Learning"
_ICLR.cc/2024/Conference — ICLR 2024 poster_

### Official Review · Reviewer_KmRA · 2023-10-23

**Soundness:** 2 fair
**Presentation:** 3 good
**Contribution:** 3 good
**Rating:** 6
**Confidence:** 3

**Summary:**

The paper proposes a new objective function for task-based continual learning (TBCL). There are two key contributions: The first one consists in highlighting that all regularisers proposed in regularised-based, memory-replay and Bayesian approaches can be seen as specific instantiations of a Bregman divergence term, thus casting the whole formulation of TBCL into a single overarching objective criterion. The second contribution consists of proposing an additional regulariser to promote generalisation, which is based on the minimisation of the L2 norm of the weighted gradient loss (here, the weight corresponds to the inverse of the Fisher Information matrix computed using the parameters from the previous tasks). A corresponding learning algorithm is proposed based on an alternating minimisation scheme. Indeed, the training algorithm updates the parameters firstly by approximately minimising the gradient loss term (the approximation avoids (i) the computation of the Hessian matrix, as assuming to be an identity, and (ii) it introduces Gaussian noise to promote exploration) and secondly by minimising the cross-entropy on the current task. Experiments on CIFAR-10, CIFAR-100 and Tiny-Imagenet demonstrate the effectiveness of the new regulariser in enhancing the generalisation performance of existing approaches. Consequently, the proposed regulariser effectively complements the ones proposed in the literature of TBCL.

**Strengths:**

1. The idea behind the two contributions is original and novel. Specifically, it is nice to see a unified view of the three classes of TBCL leveraging the general definition of Bregman divergence. Additionally, while the proposed new regulariser is known to increase generalisation in supervised deep learning, as promoting solutions characterised by flat minima [1], its adaptation to the TBCL is novel and non-trivial (**Originality**)
2. Overall, the paper is quite clear and easy to follow, except for one part (see weaknesses) (**Clarity**)
3. The solution and the experiments are convincing, demonstrating its benefits (**Significance**)

[1] Penalising Gradient Norm for Efficiently Improving Generalization in Deep Learning. ICML 2022

**Weaknesses:**

1. The main concern I have is with the overstated claim about the introduction of an innovative concept, “refresh learning”. The idea boils down to approximately minimising the weighted gradient norm, known to promote flatness in the achieved minima and therefore to increase generalisation performance. Section 3.3 is a bit handwavy and unclear. Perhaps, it is better to rephrase it directly in terms of gradient penalty as originally proposed in [1] and focus more on its extension to the TBCL setting. This would help to have a clearer understanding and intuition on why the proposed regulariser works (**Quality**).
2. It would be good to provide the ablation study for all datasets. Indeed how are the hyperparameters chosen in practice and how should practitioners choose them in general? This should help to give a better sense on how sensitive the hyperparameters are on different datasets (**Quality**)
3. All experiments are conducted on a similar family of natural images. It would be good to see how the proposed approach works on a traditional and more different MNIST-like benchmark (**Quality**)
4. Code is not available (**Reproducibility**)

**MINOR**

Some typos:
1. Section 3, remove “In this section”
2. Last paragraph page 4 (and also later in page 5) -> the NEGATIVE entropy function
3. Eq. (5) misses the expectation term in its second addend
4. Eq. (8) and all occurrences of F -> missing L

**Questions:**

All questions are related to the main weaknesses:
1. Why not directly minimising the (weighted) gradient penalty? Also, what is the advantage of introducing the Gaussian noise (it would be good to see its necessity in practice)?
2. Can you provide the complete ablation analysis on all datasets and also include some experiments about MNIST?

---

> ### Author Response · Authors · 2023-11-22
> **Response to reviewer KmRA (1/2)**
>
> Dear reviewer,
>
> We would like to thank you for your constructive feedback and sincerely appreciate your detailed suggestions.
>
> **Q1: The main concern I have is with the overstated claim about the introduction of an innovative concept, “refresh learning”. The idea boils down to approximately minimising the weighted gradient norm, known to promote flatness in the achieved minima and therefore to increase generalisation performance. Section 3.3 is a bit handwavy and unclear. Perhaps, it is better to rephrase it directly in terms of gradient penalty as originally proposed in [1] and focus more on its extension to the TBCL setting. This would help to have a clearer understanding and intuition on why the proposed regulariser works (Quality).**
>
> **A**: Thank you for your valuable suggestions.
>
>
>
>
>
>
> While interpreting our method as a weighted gradient penalty offers intuitive insights, it's essential to note that minimizing the gradient norm can be challenging due to the involved calculation of Hessian matrices, as demonstrated in the derivations below. This process can be both memory and computation inefficient. Our refresh learning method is designed to address and circumvent these challenges, leading to significantly improved memory and computation efficiency. Notably, our approach achieves these advantages *without the need to calculate the Hessian matrix*. Moreover, the introduced refresh learning concept aims to emulate the learning process of the human brain. We prioritize the retention of crucial information over exhaustive detail, reflecting a novel perspective. This perspective suggests that a more efficient learning strategy for acquiring new tasks involves preserving essential information while permitting the forgetting of less critical details from prior information. This approach facilitates a clearer understanding of our methodology for the reader.
>
>
> *gradient of gradient norm derivation*
>
> $\nabla_{\theta}||\nabla_{\theta} L^{CL}(\theta)F^{-1}|| = \nabla_{\theta}(||\nabla_{\theta} L^{CL}(\theta)F^{-1}||^2)^{\frac{1}{2}} \\
>     = \frac{1}{2} (||\nabla_{\theta} L^{CL}(\theta)F^{-1}||^2)^{-\frac{1}{2}} (2\nabla_{\theta} L^{CL}(\theta)F^{-1}) \nabla_{\theta}^2 L^{CL}(\theta)F^{-1}= \frac{\nabla_{\theta} L^{CL}(\theta)F^{-1} \nabla_{\theta}^2 L^{CL}(\theta)F^{-1}}{||\nabla_{\theta} L^{CL}(\theta)F^{-1}||}$
>
>
> $\approx \nabla_{\theta}^2 L^{CL}(\theta) F^{-1} \frac{ \nabla_{\theta} L^{CL}(\theta)}{||\nabla_{\theta} L^{CL}(\theta)||}$
>
> **Q2: It would be good to provide the ablation study for all datasets. Indeed how are the hyperparameters chosen in practice and how should practitioners choose them in general? This should help to give a better sense on how sensitive the hyperparameters are on different datasets (Quality)**
>
> **A**: Thanks for your suggestions, we provide more ablation studies on other datasets, CIFAR10 and Tiny-ImageNet, in the following tables. The results show that with an increase of $\gamma$, the performance improves slightly. Increasing the number of steps only slightly increases the performance. When unlearning with 3 steps, the performance drops due to the stronger forgetting effect.
>
>
>
> The sensitivity analysis of hyperparameters on CIFAR10 is shown in the following table:
>
> | **$\gamma$** | **0.02** | **0.03** | **0.04** |
> |--------------|-----------|-----------|-----------|
> | **Accuracy** | 94.27 $\pm$ 0.42 | 94.64 $\pm$ 0.38 | 94.82 $\pm$ 0.51 |
>
> | **$J$** | **1** | **2** | **3** |
> |---------|-------|-------|-------|
> | **Accuracy** | 94.64 $\pm$ 0.38 | 94.73 $\pm$ 0.43 | 93.50 $\pm$ 0.57 |
>
>
>
>
>
> The sensitivity analysis of hyperparameters on Tiny-ImageNet is shown in the following table:
>
> | **$\gamma$** | **0.02** | **0.03** | **0.04** |
> |--------------|-----------|-----------|-----------|
> | **Accuracy** | 53.27 $\pm$ 0.72 | 54.06 $\pm$ 0.79 | 54.21 $\pm$ 0.83 |
>
> | **$J$** | **1** | **2** | **3** |
> |---------|-------|-------|-------|
> | **Accuracy** | 54.06 $\pm$ 0.79 | 54.17 $\pm$ 0.91 | 52.29 $\pm$ 0.86 |

---

> > ### Author Response · Authors · 2023-11-22
> > **Response to reviewer KmRA (2/2)**
> >
> > **Q3: All experiments are conducted on a similar family of natural images. It would be good to see how the proposed approach works on a traditional and more different MNIST-like benchmark (Quality)**
> >
> > **A**:  Thanks for your suggestions. We perform experiments on PMNIST and RMNIST. The results are shown in the following table. The results show that our method can still outperform compared baselines by 0.5%-1.8%. Adaptive unlearning could mitigate the catastrophic interference.
> >
> >
> > | **Algorithm Method** | **P-MNIST (Domain-IL)** | **R-MNIST (Domain-IL)** |
> > |-----------------------|--------------------------|--------------------------|
> > | oEWC                  | $59.57\pm2.37$           | $77.35\pm5.77$           |
> > | oEWC+refresh          | **61.23 $\pm$ 2.18**     | **79.21 $\pm$ 4.98**     |
> > | ER                    | $78.45\pm0.72$           | $88.91\pm1.44$           |
> > | ER+refresh            | **80.28 $\pm$ 1.06**     | **90.53 $\pm$ 1.67**     |
> > | DER++                 | $88.21\pm0.39$           | $92.77\pm1.05$           |
> > | DER+++refresh         | **88.93 $\pm$ 0.58**     | **93.28 $\pm$ 0.75**     |
> >
> > **Q4: Code is not available (Reproducibility)**
> >
> > **A**:  Thanks for your suggestions. We are in the process of preparing the GitHub repository for code release. We will release code upon acceptance.
> >
> >
> >
> > **Q5: Why not directly minimize the (weighted) gradient penalty? Also, what is the advantage of introducing the Gaussian noise (it would be good to see its necessity in practice)?**
> >
> >
> > **A**:  **why not minimize weighted gradient penalty**:
> >
> >  It's essential to note that minimizing the gradient norm can be challenging due to the involved calculation of Hessian matrices, as demonstrated in the derivations below. This process can be both memory and computation inefficient. Our refresh learning method is designed to address and circumvent these challenges, leading to significantly improved memory and computation efficiency without the need to calculate the Hessian matrix.
> >
> > *gradient of gradient norm derivation*
> >
> > $\nabla_{\theta}||\nabla_{\theta} L^{CL}(\theta)F^{-1}|| = \nabla_{\theta}(||\nabla_{\theta} L^{CL}(\theta)F^{-1}||^2)^{\frac{1}{2}} \\
> >     = \frac{1}{2} (||\nabla_{\theta} L^{CL}(\theta)F^{-1}||^2)^{-\frac{1}{2}} (2\nabla_{\theta} L^{CL}(\theta)F^{-1}) \nabla_{\theta}^2 L^{CL}(\theta)F^{-1}= \frac{\nabla_{\theta} L^{CL}(\theta)F^{-1} \nabla_{\theta}^2 L^{CL}(\theta)F^{-1}}{||\nabla_{\theta} L^{CL}(\theta)F^{-1}||}$
> >
> > $\approx \nabla_{\theta}^2 L^{CL}(\theta) F^{-1} \frac{ \nabla_{\theta} L^{CL}(\theta)}{||\nabla_{\theta} L^{CL}(\theta)||}$
> >
> > **The benefits of adding Gaussian noise are two fold**:
> >
> > First, it can help to escape local minima and achieve global minimizers [1]. Second, it helps to escape the saddle point [2].  The empirical evaluations are shown as the following table:
> >
> > | **Dataset**    | **With Gaussian noise** | **Without Gaussian noise** |
> > |----------------|--------------------------|---------------------------|
> > | **CIFAR10**     | 94.64 $\pm$ 0.38        | 94.35 $\pm$ 0.31          |
> > | **CIFAR100**    | 77.71 $\pm$ 0.85        | 77.25 $\pm$ 0.96          |
> > | **Tiny-ImageNet**| 54.06 $\pm$ 0.79       | 53.82 $\pm$ 0.87          |
> >
> >
> > We can observe that with Gaussian noise, the performance can be improved further due to escaping the local minima and saddle points.
> >
> >
> >
> >
> >
> >
> > Reference:
> >
> >
> > [1] Non-Convex Learning via Stochastic Gradient Langevin Dynamics: A Nonasymptotic Analysis. COLT 2017
> >
> > [2] Escaping From Saddle Points –Online Stochastic Gradient for Tensor Decomposition. COLT 2015
> >
> >
> > **Q6:  Can you provide the complete ablation analysis on all datasets and also include some experiments about MNIST?**
> >
> >
> > **A**: Thanks for your suggestions. We would like to invite you to refer to the answers for **Q2** and  **Q3**. The results show that with an increase of $\gamma$, the performance improves slightly. Increasing the number of steps only slightly increases the performance. When unlearning with 3 steps, the performance drops due to the stronger forgetting effect. In PMNIST and RMNIST, our method also improves the performance.

---

> > > ### Author Response · Authors · 2023-11-23
> > > **Additional Discussion**
> > >
> > > Dear Reviewer KmRA,
> > >
> > >           We express our gratitude for your constructive feedback and sincerely value your detailed suggestions. As the reviewer-author discussion period is concluding shortly, we would like to inquire if there are any remaining concerns that require our clarification. Thank you!

---

### Official Review · Reviewer_tVq3 · 2023-10-29

**Soundness:** 1 poor
**Presentation:** 2 fair
**Contribution:** 2 fair
**Rating:** 3
**Confidence:** 4

**Summary:**

The paper considers a unified objective that includes prior methods as special cases.

It further proposes an unlearn-relearn method to minimize the proposed objective function.

Next, there is a short section on theoretical analysis.

Finally, experiments are conducted in comparison to prior methods.

**Strengths:**

Omitted.

**Weaknesses:**

In my opinion, the paper has the following weaknesses:
- The proposed framework is not very interesting. Basically, it says that prior work has loss functions $L_1,L_2,L_3$, therefore let us propose a unified objective $\alpha_1L_1 + \alpha_2 L_2 + \alpha_3L_3$. In my eyes this proposal is incremental and has limited novelty.


- The work is not solid and there are issues with writing. For example:
   - The first part of the paper (pages 1-5, until section 3.3), appears to be a review of prior works. This is more than half of the main paper.
   - There are many repetitive sentences. This is just one example (and there are more): There is a long paragraph on page 2 discussing refresh learning, and then a highly similar paragraph appeared on page 6.

I was not very convinced by what the paper argues about over-memorization. But after witnessing the repetitive style of the paper I realized that over-memorization is indeed harmful.

The theoretical analysis is very informal. I don't see how the theory statement connects to the proofs. The proofs seem to be written in a rush. Please justify that.

**Questions:**

I have no specific questions.

---

> ### Author Response · Authors · 2023-11-22
> **Response to reviewer tVq3 (1/3)**
>
> Dear reviewer,
>
> We would like to express our sincere gratitude for your helpful comments.
>
>
> **Q1: The proposed framework is not very interesting. Basically, it says that prior work has $L_1$, $L_2$, and $L_3$, loss functions, therefore let us propose a unified objective, $\alpha_1L_1 +\alpha_2L_2 + \alpha_3L_3$. In my eyes this proposal is incremental and has limited novelty.**
>
> **A**: We would like to emphasize that our proposed framework goes beyond a simple combination of loss functions from existing works. Its significance and non-trivial nature are rooted in three key aspects.
>
> * Firstly, our framework is broad and serves as a unifying platform for various Continual Learning (CL) methods belonging to different categories, employing Bregman divergence. With just two terms, our framework encapsulates at least six existing CL methods. Traditional methods have emerged from diverse perspectives such as regularization-based, Bayesian-based, and memory-based approaches. The absence of a general framework to unite these methods from a single viewpoint has been a gap in the field, and our proposal addresses this by offering a unified optimization objective.
>
> * Secondly, this unified framework provides a more profound understanding of existing CL methods. It not only highlights the shared characteristics among these methods but also exposes their limitations, particularly in the context of over-memorization. However, it's crucial to emphasize an important insight revealed from our framework: the emphasis on remembering essential information rather than every detail. This nuanced perspective suggests that a more effective learning strategy for acquiring new tasks involves retaining essential information while allowing for the forgetting of less critical details from old information. By uncovering these insights, the framework contributes to a more comprehensive comprehension of the CL landscape.
>
> * Thirdly, leveraging the insights provided by our framework, it becomes a general guideline for the development of novel CL methods. For instance, our proposed refresh learning is a direct derivation from this unified framework, as highlighted in our current revision.
>
> In summary, the contribution of our framework is substantial and goes beyond mere aggregation. It establishes a unified perspective for diverse CL methods, deepens our understanding of existing approaches, and serves as a guideline and catalyst for the development of novel CL methods.

---

> ### Author Response · Authors · 2023-11-22
> **Response to reviewer tVq3 (2/3)**
>
> **Q2: The work is not solid and there are issues with writing. For example:
> The first part of the paper (pages 1-5, until section 3.3), appears to be a review of prior works. This is more than half of the main paper.
> There are many repetitive sentences. This is just one example (and there are more): There is a long paragraph on page 2 discussing refresh learning, and then a highly similar paragraph appeared on page 6.
> I was not very convinced by what the paper argues about over-memorization. But after witnessing the repetitive style of the paper I realized that over-memorization is indeed harmful.**
>
>
> **A**: Thank you for your valuable suggestions.
>
>
> **Our work is actually solid for three reasons.**
>
> * Firstly, it consolidates existing methods from diverse categories into a cohesive framework, constituting a noteworthy contribution to CL. Additionally, we offer comprehensive math derivations.
>
> * Secondly, our unified framework enhances the comprehension of established CL methods, shedding light on their limitations. Furthermore, the insights offered by our framework serve as motivation for the development of novel CL methods.
>
> * Thirdly, from our framework, we derive a novel refresh learning approach, showcasing the superior and widely applicable nature of our framework.
>
> **presentation of the first part, page 1-5**: In response, page 1-3 is the abstract，introduction and related work, which are necessary parts of the paper. Page 4-5 is to unify existing methods. We use the minimal descriptions to be self-contained to describe existing methods and how our proposed framework corresponds to different existing CL methods. We have condensed the first part of the main paper to be less than two pages. Furthermore, the proposed framework in the first part reveals the limitations in existing CL methods, and our proposed new CL method (refresh learning) is derived and motivated from the framework proposed in the first part.  These two parts are constituted as a whole.
>
>
> **repetitive sentences**. We write repetitive sentences to emphasize the issue of over-memorization. To address concerns about repetitiveness, we have streamlined and minimized redundant sentences that emphasize over-memorization. In addition, we provide a concrete example of human learning in the introduction to illustrate our idea. We also delete other repetitive sentences in this revision. We invite you to explore our revised introduction on page 2, which is highlighted in blue, for a more focused presentation. Below, for better readability, we present the revised version in the following:
>
> On one hand, forgetting can be beneficial for the human brain in various situations, as it helps in efficient information processing and decision-making. One example is the phenomenon known as "cognitive load".
> Imagine a person navigating through a new big city for the first time. They encounter a multitude of new and potentially overwhelming information, such as street names, landmarks, and various details about the environment. If the brain were to retain all this information indefinitely, it could lead to cognitive overload, making it challenging to focus on important aspects and make decisions effectively.
> However, the ability to forget less relevant details allows the brain to prioritize and retain essential information. Over time, the person might remember key routes, important landmarks, and necessary information for future navigation, while discarding less critical details. This selective forgetting enables the brain to streamline the information it holds, making cognitive processes more efficient and effective.
> In this way, forgetting serves as a natural filter, helping individuals focus on the most pertinent information and adapt to new situations without being overwhelmed by an excess of irrelevant details. On the other hand, CL involves adapting to new tasks and acquiring new knowledge over time. If a neural network remembers every detail from all previous tasks, it could quickly become impractical and resource-intensive. Forgetting less relevant information helps in managing memory resources efficiently, allowing the model to focus on the most pertinent knowledge. Furthermore, catastrophic interference occurs when learning new information disrupts previously learned knowledge. Forgetting less relevant details helps mitigate this interference, enabling the model to adapt to new tasks without severely impacting its performance on previously learned tasks.

---

> > ### Author Response · Authors · 2023-11-22
> > **Response to reviewer tVq3 (3/3)**
> >
> > **Q3: The theoretical analysis is very informal. I don't see how the theory statement connects to the proofs. The proofs seem to be written in a rush. Please justify that.**
> >
> > **A**:   Thanks for your question. We outline our proof in the following. We denote the weighted gradient norm regularized CL loss function as $L_{GN}(\theta)$ and our refresh learning loss function as $L_{refresh}(\theta)$. Then, we calculate their gradient $\nabla_{\theta}L_{GN}(\theta)$ and $\nabla_{\theta}L_{refresh}(\theta)$, respectively. Finally, we show their gradient is approximately the same, i.e., $\nabla_{\theta}L_{GN}(\theta) \approx \nabla_{\theta}L_{refresh}(\theta)$, then the conclusion follows. We invite you to refer to the **new proof details in Appendix B**, highlighted in blue.

---

> > > ### Author Response · Authors · 2023-11-23
> > > **Request Discussion**
> > >
> > > Dear Reviewer tVq3,
> > >
> > >          We appreciate your helpful suggestions. As the reviewer-author discussion period is concluding soon, we would like to inquire if there are any remaining concerns that require additional clarification. Thank you!

---

### Official Review · Reviewer_Dw6m · 2023-10-31

**Soundness:** 3 good
**Presentation:** 3 good
**Contribution:** 3 good
**Rating:** 6
**Confidence:** 2

**Summary:**

This paper proposes a unified optimization objective which is capable of encompassing existing CL approaches, including regularization based/Bayesian-based and memory-replay based methods. From the objective, the authors identified a novel method of refresh-learning, which act as a plug-in to augment the performance of CL methods.

**Strengths:**

The paper presents a unified framework for different continual learning methods, and derived a new CL approach from the unified objective. The paper presents detailed theoretical derivations of the algorithm and comprehensive experimental results to demonstrate the advantage of the new method of refresh learning.

**Weaknesses:**

For me the more interesting and novel part of the paper is the refresh learning method, its derivation, intuition behind it, and its performance, while the part where the unified approach corresponds to different CL methods in different setup is more expected and easier to follow. I would recommend the authors shorten the part of how the unified objective corresponds to different special cases and further elaborate on refresh learning.

**Questions:**

1.Why can't the over-memorization issue be solved by properly tuning the regularization strengths alpha/beta?
2.Can you provide an intuitive explanation why unlearning the posterior first and relearning a single set of parameters from the unlearned posterior is a good method for encouraging forgetting of outdated information?

---

> ### Author Response · Authors · 2023-11-22
> **Response to reviewer Dw6m**
>
> Dear reviewer,
>
> We would like to express our sincere appreciation for your valuable suggestions.
>
>
> **Q1: For me the more interesting and novel part of the paper is the refresh learning method, its derivation, intuition behind it, and its performance, while the part where the unified approach corresponds to different CL methods in different setup is more expected and easier to follow. I would recommend the authors shorten the part of how the unified objective corresponds to different special cases and further elaborate on refresh learning.**
>
> **A**:  Thank you for your suggestions. In this revised version, we have condensed the section discussing how the unified objective aligns with various special cases to be under two pages. Additionally, we have expanded on the details and explanations regarding refresh learning.
>
>
> **Q2: Why can't the over-memorization issue be solved by properly tuning the regularization strengths alpha/beta? 2.Can you provide an intuitive explanation why unlearning the posterior first and relearning a single set of parameters from the unlearned posterior is a good method for encouraging forgetting of outdated information?**
>
>
> **A**: Thank you for your question!
>
> **tuning $\alpha, \beta$ for over-memorization**: Given the dynamic nature of the CL process, determining the appropriate regularization strengths ($\alpha$ and $\beta$) poses a challenge, as they need to be adjusted on-the-fly. The difficulty is centered around dynamically establishing the strengths rather than predefining them. Moreover, a constant regularization ratio $\alpha/\beta$ is insufficient for capturing the varied importance of parameters, especially when dealing with a large number of parameters in the backbone since each parameter is associated with different importance for the previous tasks.
>
> In contrast, our approach excels at adaptively determining the degree of content to be forgotten. Our method discerns the importance of *individual parameters*, allowing for a nuanced approach where certain parameters crucial for retaining knowledge incur a substantial penalty, while less critical parameters face a smaller penalty.
>
> It's worth noting that $\alpha$ and $\beta$ alone govern the overall speed and rate of forgetting or remembering across the entire network, lacking the granularity to determine specific content that should be retained or forgotten and the respective speed for each parameter. We delve into these considerations in the following equation, which shows that $\alpha$ and $\beta$ can only determine the speed of remembering for the **entire network**, but could not adaptively adjust the unlearning speed for **each neural network parameter**. More derivation details are provided in **Appendix A.6**.
>
> \begin{equation}
>     \theta_{k+1} =  \theta_k -((\alpha F +
>  \beta I)^{-1} \nabla_{\theta}L_{CE}(\theta_k) )
> \end{equation}
>
> **Intuitive explanations for unlearning and relearning:** Unlearning the posterior first and subsequently relearning a single set of parameters from the unlearned posterior is an effective method for encouraging forgetting of outdated information. This approach is intuitive because it mimics a process of deliberate forgetting in the human learning paradigm.
>
> When we unlearn the posterior, we essentially prioritize the removal or down-weighting of information related to outdated tasks or experiences. By resetting the parameters to a state derived from this unlearned posterior, we free up the model parameter space to adapt to new information. This process encourages the model to focus on the most relevant knowledge, essentially discarding the less pertinent details from the past.
>
> In essence, unlearning followed by relearning allows the model to shed the influence of outdated information, promoting adaptability and preventing interference from irrelevant details. It facilitates a more efficient and targeted learning experience, akin to how humans naturally prioritize relevant and essential information over less pertinent knowledge.

---

> > ### Author Response · Authors · 2023-11-23
> > **Further Discussion**
> >
> > Dear reviewer Dw6m,
> >
> >            We genuinely appreciate your valuable suggestions. We are curious if there are any aspects that may require further clarification. We could provide further response. Thank you!

---

### Official Review · Reviewer_ngze · 2023-11-08

**Soundness:** 3 good
**Presentation:** 2 fair
**Contribution:** 2 fair
**Rating:** 6
**Confidence:** 3

**Summary:**

In this paper, the authors provide a unified framework for various types of continual learning (CL) algorithms, Specifically, by using Bregman divergence, they show that well-known CL approaches can be viewed as instances of the general objective coming from different parametrization of the Bregman divergence function. Then, they propose a new CL algorithm, which consists of a two-step process: unlearn and relearn. The authors provide empirical results that show the merits of their approach as compared to other relevant CL algorithms.

**Strengths:**

- The unified framework provided by the authors is interesting and sheds light on particular characteristics of existing algorithms.
- The main idea behind the proposed "refresh learning" algorithm seems to be reasonable.
- The authors integrated their "refresh learning" approach to existing algorithms showing empirical results on three different datasets.

**Weaknesses:**

- The first part of the paper (unified framework) seems to be unrelated to the second one (Refresh learning algorithm).
- The refresh learning objective eqs (12), (13) is insufficiently motivated.
- It's not clear why unlearning should be enforced on the current batch. The authors do not provide any motivation.
- The conversion of the constraint to a PDE is not obvious to me and is not sufficiently explained in the paper.
- The refresh learning algorithm ends up in a preconditioned ascent followed by a decent step. But the descent step seems to be applied in a different loss (??) (eq. 6 in Algorithm 1). This is a bit confusing.
- The authors integrate the proposed algorithm into various existing schemes. However, this is arbitrarily introduced in the experimental section only, and not clear how this integration can naturally arise from the problem introduced in section 3.3 or the unified CL framework of section 3.2.

** Post-rebuttal comment: I appreciate the authors' efforts to address my concerns. The revised version of the paper has now been improved. The proposed refresh learning framework is better motivated and the derivations provided by the authors make the paper easy to follow. Therefore, I have decided to increase my score.

**Questions:**

- Could the proposed refresh learning algorithm be derived by the unified CL framework?
- Why energy functional is defined as in (13)? Could you please provide more intuition? Are there any other ways to promote unlearning?
- What is the loss function in eq. (6) of Algorithm 1?

---

> ### Author Response · Authors · 2023-11-22
> **Response to reviewer ngze (1/3)**
>
> Dear reviewer,
>
> We sincerely appreciate your thoughtful feedback on our work.
>
>
> **Q1: The first part of the paper (unified framework) seems to be unrelated to the second one (Refresh learning algorithm).**
>
> **A**: Thanks for pointing out this! In fact, the second part (refresh learning) is closely related to the first part (unified framework) for two reasons.
>
> * First, the second part is built on top of the first part. This unified framework (first part) provides a deeper understanding of existing CL methods. It not only highlights the shared characteristics among these methods but also exposes their limitations, particularly in the context of over-memorization. It's crucial to emphasize an important insight revealed from our framework: the emphasis on remembering essential information rather than every detail. In other words, the first part provides motivation and insight for the second part (our method).
>
> * Second, leveraging the insights provided by our framework (first part), the proposed unified framework becomes a general guideline for the development of novel CL methods (second part). For instance, our proposed refresh learning is a direct derivation from this unified framework, as highlighted in our current revision. Further, the algorithm 1 is applied on the general CL loss function proposed in the first part. We invite you to refer to the method section highlighted in blue to see the interconnections between those two sections.
>
>
>
>
>
> **Q2: The refresh learning objective eqs (12), (13) is insufficiently motivated**
>
> **A**:  Thanks for pointing out this. We will derive these learning objectives below to show their motivation. The main objective is to minimize the KL divergence between the current CL model parameters posterior and the target unlearned model parameter posterior. We denote the CL model parameter posterior at time $t$ as $\rho_t$, the target unlearned posterior as $\mu$. The goal is to minimize $KL(\rho_t||\mu)$. Following [1], we define the target unlearned posterior as a energy function $\mu = e^{-\omega}$ and $\omega = -L^{CL}$.
> This KL divergence can be further decomposed as
>
> $KL(\rho_t||\mu) = \int \rho_t(\theta) \log \frac{\rho_t(\theta)}{\mu(\theta)} d\theta = - \int \rho_t(\theta) \log \mu(\theta) d\theta + \int \rho_t(\theta) \log \rho_t(\theta) d\theta = H(\rho_t, \mu) - H(\rho_t)$
>
>
> $H(\rho_t, \mu):= -E_{\rho_t} \log \mu$ is the cross-entropy between $\rho_t$ and $\mu$. $H(\rho_t) := -E_{\rho_t} \log \rho_t$ is the entropy of $\rho_t$.
>
>  Then, we plugin these terms into the above equation, and obtain the following:
>
> $KL(\rho_t||\mu) = -E_{\rho_t} \log \mu + E_{\rho_t} \log \rho_t = - E_{\rho_t} L^{CL} + E_{\rho_t} \log \rho_t$
>
> The above equation is precisely our refresh learning objective.
>
> We would like to invite you to refer to the method section highlighted in blue (eq (9) to eq (12) in the main text).
>
> **Q3: It's not clear why unlearning should be enforced on the current batch. The authors do not provide any motivation.**
>
> **A**: Thanks for pointing out this. In our discussion of the "current batch," we specifically refer to the batch of data available for use during the CL process. This concept is applicable within the broader context of CL scenarios. In memory-based CL methods, access to a batch of data extends to both the current task and the memory buffer. In such cases, the "current batch data" encompasses data from both the current new task and the stored memory. Conversely, CL methods falling into other categories have access solely to the batch of data associated with the current task and do not leverage information from the memory buffer.
>
> **Reference:**
>
> [1] Sampling as optimization in the space of measures: The Langevin dynamics as a composite optimization problem. COLT 2018

---

> ### Author Response · Authors · 2023-11-22
> **Response to reviewer ngze (2/3)**
>
> **Q4: The conversion of the constraint to a PDE is not obvious to me and is not sufficiently explained in the paper.**
>
>
> **A**: Thank you for bringing this to our attention. We derive the PDE as below
>
> By Fokker-Planck equation, the gradient flow of KL divergence, i.e. $KL(\rho_t||\mu)$ is as following:
> \begin{equation}
>     \frac{\partial\rho_t}{\partial t} = div\left(\rho_t \nabla \frac{\delta KL(\rho_t||\mu) }{\delta \rho}(\rho)\right)
> \end{equation}
>  $div \cdot (q) : = \sum_{i=1}^{d}\partial_{z^{i}} q^{i}(z)$ is the divergence operator operated on a vector-valued function $q: R^d \to R^d$, where $z^{i}$ and $q^{i}$ are the $i$ th element of $z$ and $q$.
> Then, since the first-variation of KL-divergence, i.e., $\frac{\delta KL(\rho_t||\mu) }{\delta \rho}(\rho_t) = \log\frac{\rho_t}{\mu} + 1$. We plug this equation into the above equation, and obtain the following:
> \begin{equation}
>      \frac{\partial\rho_t(\theta)}{\partial t} = div (\rho_t(\theta)\nabla(\log\frac{\rho_t(\theta)}{\mu} + 1)) = div (\nabla \rho_t(\theta) + \rho_t(\theta)\nabla \omega)
> \end{equation}
> Then, [2] proposes a more general Fokker-Planck equation as following by adding two additional matrix:
> \begin{equation}
>       \frac{\partial\rho_t(\theta)}{\partial t} = div [([D(\theta) + Q(\theta)])(\nabla \rho_t(\theta) + \rho_t(\theta)\nabla \omega)]
> \end{equation}
> where $D(\theta)$ is a positive semidefinite matrix and $Q(\theta)$ is a skew-symmetric matrix.
> We plug in the defined $\omega = -L^{CL}$ into the above equation, we can get the following PDE:
> \begin{equation}
>         \frac{\partial\rho_t(\theta)}{\partial t} = div ([D(\theta) + Q(\theta)])[-\rho_t(\theta)\nabla  L^{CL}(\theta) + \nabla \rho_t(\theta)]
> \end{equation}
>
> The above PDE is precisely what we present in our paper.
>
> In this revised version, we have taken care to offer additional details and derivations of this partial differential equation (PDE), expanding on the content from Eq. (13) to Eq. (16) in the main text.
>
>
> **Q5:  The refresh learning algorithm ends up in a preconditioned ascent followed by a decent step. But the descent step seems to be applied in a different loss (??) (eq. 6 in Algorithm 1). This is a bit confusing.**
>
> **A**: Thanks for pointing out this!  The loss function in eq. 6 is the same as that $L^{CL}$. This is a typo and we have corrected this in our current revision.
>
> **Q6:  The authors integrate the proposed algorithm into various existing schemes. However, this is arbitrarily introduced in the experimental section only, and not clear how this integration can naturally arise from the problem introduced in section 3.3 or the unified CL framework of section 3.2.**
>
> **A**: Thanks for your question. We apply the refresh learning method on the general CL loss function introduced in Section 3.2. For example, for DER, we use the DER loss as the $L_{CL}$. Then, we apply the method in Algorithm 1 to perform the refresh learning. The integration with other methods are similar.
>
> **Reference**:
>
> [2] A Complete Recipe for Stochastic Gradient MCMC. Neurips 2015

---

> ### Author Response · Authors · 2023-11-22
> **Response to reviewer ngze (3/3)**
>
> **Q7: Could the proposed refresh learning algorithm be derived by the unified CL framework?**
>
> **A**:  Yes, we can derive the refresh learning from the unified framework as below.
>
> **Refresh Learning As a Special Case**
> Now, we derive refresh learning as a special case of our unified framework:
>
> $L_{unlearn}  =   L_{CE}(x, y) + 2\alpha D_{\Phi}(h_{\theta}(x), z) + \beta D_{\Psi}(\theta, \theta_{old}) -\alpha D_{\Phi}(h_{\theta}(x), z)$
>
> In the above equation, we adopt the second-order Taylor expansion on $D_{\Phi}(h_{\theta}(x), z)$ as the following:
>
> $D_{\Phi}(h_{\theta}(x), z)  \approx  D_{\Phi}(h_{\theta_{k}}(x), z) +  \nabla_{\theta} D_{\Phi}(h_{\theta_{k}}(x), z)(\theta - \theta_{k}) + \frac{1}{2} (\theta - \theta_{k})^TF(\theta - \theta_{k})$
>
>
>
> Since $\nabla_{\theta} D_{\Phi}(h_{\theta}(x), z)$ is close to zero at the stationary point, i.e.,  $\theta_{k}$, we thus only need to optimize the leading quadratic term in the above equation.
>
> We define $L_{CL} = L_{CE}(x, y) + 2\alpha D_{\Phi}(h_{\theta}(x), z) + \beta D_{\Psi}(\theta, \theta_{old}) $, then $L^{refresh} = L_{CL} - \alpha D_{\Phi}(h_{\theta}(x), z)$
>
> Further, we adopt the first-order Taylor expansion on $L_{CL}$ as the following:
> \begin{equation}
>     L_{CL}(\theta) \approx L_{CL}(\theta_{k}) +  \nabla_{\theta}L_{CL}(\theta_{k}) (\theta - \theta_{k})
> \end{equation}
> In summary, the approximate loss function for $L_{unlearn}$ can be expressed as the following:
>
> $L_{unlearn} \approx  \nabla_{\theta} L_{CL}(\theta_{k})  (\theta - \theta_{k})  - \frac{\alpha}{2} (\theta - \theta_{k})^TF(\theta - \theta_{k})$
>
> We then take the gradient with respect to $\theta$ for the right hand side of the above equation, we can obtain the following:
> \begin{equation}
> \nabla_{\theta}L_{CL}(\theta_{k})  - \alpha F(\theta - \theta_{k}) = 0
> \end{equation}
> Solving the above equation leads to the following unlearning for the previously learned tasks:
>
> $ \theta_{k}^{\prime} =  \theta_k +\frac{1}{\alpha} F^{-1} \nabla_{\theta}L_{CL}(\theta_{k}) $
>
> The above equation is nearly identical to our refresh learning update equation, as shown below:
>
>
>  $\theta^{k} = \theta^{k-1} + \gamma[F^{-1}\nabla L^{CL}(\theta^{k-1})] + N (0, 2\gamma F^{-1})$
>
> with the only distinction being that our refresh learning update (above equation) incorporates an additional random noise perturbation. This helps the CL model escape local minima  and saddle point. The constant $\frac{1}{\alpha}$ now takes on a new interpretation, serving as the unlearning rate.
>
>
> We also provide the derivation details as Eq (18) to Eq (22), which also be shown in the main text highlighted in blue.
>
>
> **Q8: Why energy functional is defined as in (13)? Could you please provide more intuition? Are there any other ways to promote unlearning?**
>
> **A**:  Thank you for your question.
>
> **why energy functional defined in current version**
>
> In the following, we derive the energy functional as below. The main objective is to minimize the KL divergence between the current CL model parameters posterior and the target unlearned model parameter posterior. We denote the CL model parameter posterior at time $t$ as $\rho_t$, the target unlearned posterior as $\mu$. The goal is to minimize $KL(\rho_t||\mu)$. Following \cite{wibisono2018sampling}, we define the target unlearned posterior as a energy function $\mu = e^{-\omega}$ and $\omega = -L^{CL}$.
> This KL divergence can be further decomposed as
>
> $KL(\rho_t||\mu) = \int \rho_t(\theta) \log \frac{\rho_t(\theta)}{\mu(\theta)} d\theta = - \int \rho_t(\theta) \log \mu(\theta) d\theta + \int \rho_t(\theta) \log \rho_t(\theta) d\theta = H(\rho_t, \mu) - H(\rho_t)$
>
>
> $H(\rho_t, \mu):= -E_{\rho_t} \log \mu$ is the cross-entropy between $\rho_t$ and $\mu$. $H(\rho_t) := -E_{\rho_t} \log \rho_t$ is the entropy of $\rho_t$. Then, we plugin these terms into the above equation, and obtain the following:
>
> $KL(\rho_t||\mu) = -E_{\rho_t} \log \mu + E_{\rho_t} \log \rho_t = - E_{\rho_t} L^{CL} + E_{\rho_t} \log \rho_t$
>
> The above equation is precisely the energy functional we aims to optimize.
>
>  In this updated version, we offer further derivations and insights, spanning from Eq. (9) to Eq. (10) in the main text highlighted in blue with the same text as above.
>
>
> **other ways to promote unlearning**: One of other ways to promote unlearning is to enforce the unlearned posterior distribution to target posterior distribution by $\chi^2$ divergence. In other words, the goal is to minimize $\chi^2(\rho_t||\mu)$. To obtain the unlearned posterior, we solve the following Fokker-Planck equation:
>
> $\frac{\partial\rho(\theta)}{\partial t} = div (\nabla (\frac{\rho(\theta)}{\mu}))$.
>
>
>
>
> **Q9: What is the loss function in eq. (6) of Algorithm 1?**
>
> **A**: It is the same as $L_{CL}$.

---

> > ### Author Response · Authors · 2023-11-23
> > **Require Discussion**
> >
> > Dear Reviewer ngze,
> >
> >         We sincerely appreciate your constructive feedback and have made every effort to address your concerns in our rebuttal. Since the reviewer-author discussion period will be closing in a few hours, we would like to confirm if our response resolves the issues you raised. Thank you!

---

### Meta-Review · Area_Chair_eSVK · 2023-12-18

**Metareview:**

This paper proposes a unified view of CL that encompasses several methods for solving the CL problem. Based on this framework, the authors also propose a new refresh-learning mechanism for addressing forgetting. Their solution is complementary to existing methods like ER and when combined, their method achieves better performance.

In general, the authors addressed all the concerns of the reviewers during the rebuttal. The only negative criticism is my reviewer tVq3 but I am ignoring it since the complaint about novelty is not valid. This work advances the knowledge in the field of CL.

Overall, this is a solid work that deserves acceptance.

**Justification For Why Not Higher Score:**

This is a solid work that is at the poster level.

**Justification For Why Not Lower Score:**

No reason to reject this paper.

---

### Decision · Program_Chairs · 2024-01-16

Accept (poster)